# Micromachined structures decoupling Joule heating and electron wind force

**Shaojie Gu** [1] ✉, **Yasuhiro Kimura** [1], **Xinming Yan** [1], **Chang Liu** [1], **Yi Cui** [2], **Yang Ju** [3] ✉ **& Yuhki Toku** [1] ✉

Microstructural changes in conductive materials induced by electric current treatments, such as electromigration and electroplasticity, are critical in semiconductor and metal processing. However, owing to the inevitable thermal effect (Joule heating), the athermal effect on microstructural modifications remains obscure. This paper presents an approach of utilizing pre-micromachined structures, which obstruct current flow but maintain a thermal history similar to that of the matrix, effectively disentangling the thermal and athermal effects. A duplex stainless-steel material is selected to validate the feasibility of this method. Microstructural characterizations show that the athermal effect, especially the electron wind force (EWF), primarily governs the element diffusion and phase transformation in this study. Moreover, many σ phases (Cr-enriched) are precipitated in the micromachined structures, whereas no precipitation occurred in the matrix, suggesting that the directional EWF disrupts the Cr aggregation caused by Joule heating. Furthermore, we present a critical formula for determining the dimensions of micromachined structures of commonly used metallic materials. The proposed method may serve as an effective and powerful tool for unveiling the athermal effect on microstructural alterations.

The effect of electric current treatment (ECT) on conductive materials, namely, electromigration, was first observed by the French physicist Geradin while studying liquid alloys[1]. Electromigration is caused by electron motion within conductive materials; the moving electrons transfer their kinetic energy to metal atoms, causing atomic diffusion. Additionally, when metallic materials are subjected to ECT during deformation, a reversible plastic deformation with a drop in flow stress occurs, known as electroplasticity[2–5], which can be attributed to the dislocation movements caused by the impact of electrons[2]. Electroplasticity, which softens materials and improves their ductility, is extensively applied in metal forming[6]. Moreover, ECT can also induce large-scale microstructural modifications, including rapid recrystallization[7–12], phase transformation[13–18], and damage healing[19–25].

However, engineered metallic materials experience Joule heating due to the flow of electric current. Therefore, the ECT-assisted microstructural modifications are often attributed to thermal and athermal effects[26–28]. The thermal effect primarily involves Joule heating, whereas the athermal effect follows a complex mechanism. Theoretical and experimental evidences pin the electron wind force (EWF) as the origin of the athermal effect in electromigration (atomic diffusion)[29–31]. However, the principles of electroplasticity (dislocation motion behavior) have not yet been clarified. Initially, Troitskii et al. attributed the motion of electro-induced dislocations to the EWF[3,4]. Some recent studies have also shown that the EWF plays a decisive role in dislocation motion and reconfiguration[32,33]. However, the quantified value of the EWF is insufficient to drive dislocation movement, and other causes, such as phonon wind force and thermal gradient or

[1]Department of Micro-Nano Mechanical Science and Engineering, Graduate School of Engineering, Nagoya University, Nagoya 464-8601, Japan. [2]Department of Mechanical Systems Engineering, Graduate School of Engineering, Nagoya University, Nagoya 464-8601, Japan. [3]State Key Laboratory of Fluid Power and Mechatronic Systems, School of Mechanical Engineering, Zhejiang University, Hangzhou 310030, China. ✉e-mail: gu.shaojie.e7@f.mail.nagoya-u.ac.jp; yang.ju@zju.edu.cn; toku@nagoya-u.jp

compressive stress, have been proposed[26,27]. Recent studies suggest that because different materials exhibit different electroplastic phenomena, the mechanisms of the athermal effect underlying this phenomenon might not have a unified explanation[26–28].

Moreover, because the thermal effect often accompanies the athermal effect, determining the mechanism underlying the athermal effect-induced modification of microstructures is challenging. Therefore, decoupling the thermal and athermal effects, as well as examining the role of the athermal effect in the microstructural modification, are crucial. Several approaches have been adopted to distinguish thermal and athermal effects. Some researchers have utilized pure metals with good electrical conductivity to inhibit the increase in the temperature[2–4,7,34–36]. In a few studies, experimental samples have been immersed in liquid nitrogen[3,27,36] or massive heat sinks[37] have been utilized to inhibit temperature rise. However, these approaches cannot completely inhibit the increase in the temperature. Because the thermal effect is unavoidable, some studies have employed conventional heat treatments to simulate similar thermal histories of the current-treated cases and reveal the contribution of the athermal effect[10,12,26,32,33,38–42]. However, the conventional heat treatment methods cannot achieve a high heating rate of up to $10^6 \, °C \, s^{-1}$, which can be easily achieved by a high-energy pulsed current treatment[8,43]. Additionally, in situ transmission electron microscopy (TEM) techniques provide deep insights into the mechanisms responsible for ECT-induced microstructural changes in materials, particularly the effects of dislocation motion[28,44–48]. Moreover, due to the small size of TEM samples (micro-sizes with approximately 100 nm in thickness), rapid thermal dissipation can effectively suppress the temperature elevation[28,44]. However, extensively exposed free surfaces of such ultra-thin samples can significantly influence the ECT-induced microstructural alteration of these samples (for instance, these free surfaces affect the dislocation motion[28]), thus presenting challenges in uncovering the underlying mechanisms. To date, no method or technique that can effectively distinguish and decouple thermal and athermal effects, especially quantify the role of athermal effect in material microstructural modifications, has been reported.

Here, we propose a method of using pre-micromachined structures to decouple thermal and athermal effects. The micromachined structures achieve in situ heating during electric current application, maintaining thermal histories and rapid heating rates nearly identical to those of the matrix; this feature of maintaining similar thermal histories and heating rates is unattainable by conventional heat treatment methods. We can distinguish thermal and athermal effects using this in situ heating method, which serves as an effective tool for decoupling these effects and especially investigating the athermal effect on microstructural modifications. Furthermore, a completely annealed duplex stainless-steel (DSS) material is utilized in this study. The microstructure of the DSS contains approximately 50% α-ferrite and 50% γ-austenite phases with compositional differences in phases that Cr concentrates in the α phase and Ni, Mn, and Cu enrich in the γ phase. Moreover, this DSS contains zero or negligible dislocations. Therefore, the selected material simplifies the athermal effect on the EWF (element diffusion-related), and whether the athermal effect other than the EWF influences element diffusion and phase transitions will not be discussed in this study due to the lack of experimental and theoretical evidence. Accordingly, the analyzed mechanisms of thermal and athermal effects can be attributed to Joule heating and the EWF, respectively.

## Results
### Design of the experimental procedure
Some specific structures, such as the micro-pillar and micro-cantilever shown in Fig. 1, were pre-processed on the samples using the focused ion beam (FIB) method (Supplementary Fig. 1). These micromachined structures can block the current flow while permitting heat conduction. Figure 1c, f show that heat can transfer from the matrix through the base to the micromachined pillar and cantilever. Consequently, the micromachined structures are solely influenced by the thermal effect during ECT (A' and C' regions), whereas the matrix experiences thermal and athermal effects simultaneously (B' and D' regions).

The thermal and athermal effects were quantified and distinguished by comparing the modified microstructures before and after the ECT. In situ electron backscatter diffraction (EBSD) and energy dispersive X-ray spectroscopy (EDS) analyses were performed on the micro-pillar before and after the ECT (Fig. 1b–d). The micromachined cantilever was further processed using a FIB method to prepare samples for TEM, EDS, and transmission Kikuchi diffraction (TKD) analyses (Fig. 1g, h).

The dimensions of these micromachined structures are critical; if the dimensions are large, then a long heat transfer path may prevent the micromachined structures from attaining a thermal history similar to that of the matrix region; in contrast, if the dimensions are small, then the observable region may be limited, posing challenges for the subsequent analyses. Therefore, maximizing the dimensions of the micromachined structures while ensuring that their temperature is not significantly lower than that of the matrix is essential. Additionally, the cost and feasibility of micromachining should be considered.

### Determining the critical dimension of the micromachined pillar
Figure 2a depicts the micro-pillar (with a height $H$ and length or width $L$) fabricated in the central area of the sample. Figure 2b, c show the schematics of the current blocking effect and heat conduction of the micro-pillar under ECT. In addition, we established a two-dimensional finite element (FE) model of the pillar, whose detailed governing equations, boundary conditions, and initial conditions are presented in Supplementary Note 1. The temperature-dependent physical and electrical properties of a lean DSS material[49,50] were used in the simulations to determine the critical dimension of the micro-pillar.

The analysis results demonstrated that the current blocking effect was geometry-dependent and correlated with the $H/L$ ratio of the pillar. Consequently, we simulated the current density distribution maps for pillars with different $H/L$ values under pulsed ECT at a current density of 700 A mm$^{-2}$ for 8 ms (Fig. 2d). Furthermore, the relationship between the maximum current density at the top of the pillars (Node 1) and the $H/L$ ratio in Fig. 2e shows that the current can be effectively blocked (blocking rate is larger than 98%) when $H/L \geq 1.5$.

Subsequently, we simulated the temperature distribution maps of the pillars with different $H$ (or $L$) values (Fig. 2f). The relationship between the temperature difference at the pillar top and $H$ (Fig. 2g) indicated that for $H \leq 52.5 \, \mu m$, the maximum temperature at the top of the pillar was nearly identical to that of the matrix region (difference rate <2%). However, the temperature linearly decreased when $H$ exceeded 52.5 μm. Therefore, for the DSS material used in this study, the maximum allowable dimensions for the pillar were $H \times L \times L = 52.5 \times 35 \times 35 \, \mu m^3$. The current density–time and temperature–time curves of the matrix region and pillar top under the ECT at a current density of 700 A mm$^{-2}$ for 8 ms are shown in Fig. 2h, i, respectively. The maximum current density at the pillar top was approximately 13 A mm$^{-2}$, which was only 1.9% of that in the matrix region (700 A mm$^{-2}$; Fig. 2h), indicating that the current was effectively blocked. However, the temperature–time profile at the pillar top was nearly identical to that of the matrix (Fig. 2i), with the maximum temperature reaching 629 °C, differing by only 1.7% from the matrix (640 °C). Hence, the pillar top region was predominantly affected only by the thermal effect, while the matrix region was subjected to both thermal and athermal effects.

### Determining the size of the micromachined cantilever
Figure 3 presents the simulation results for the micromachined cantilever to determine its dimension. Figure 3a illustrates the schematic of the micromachined cantilever (length $H$, height $L$, and thickness $T$)

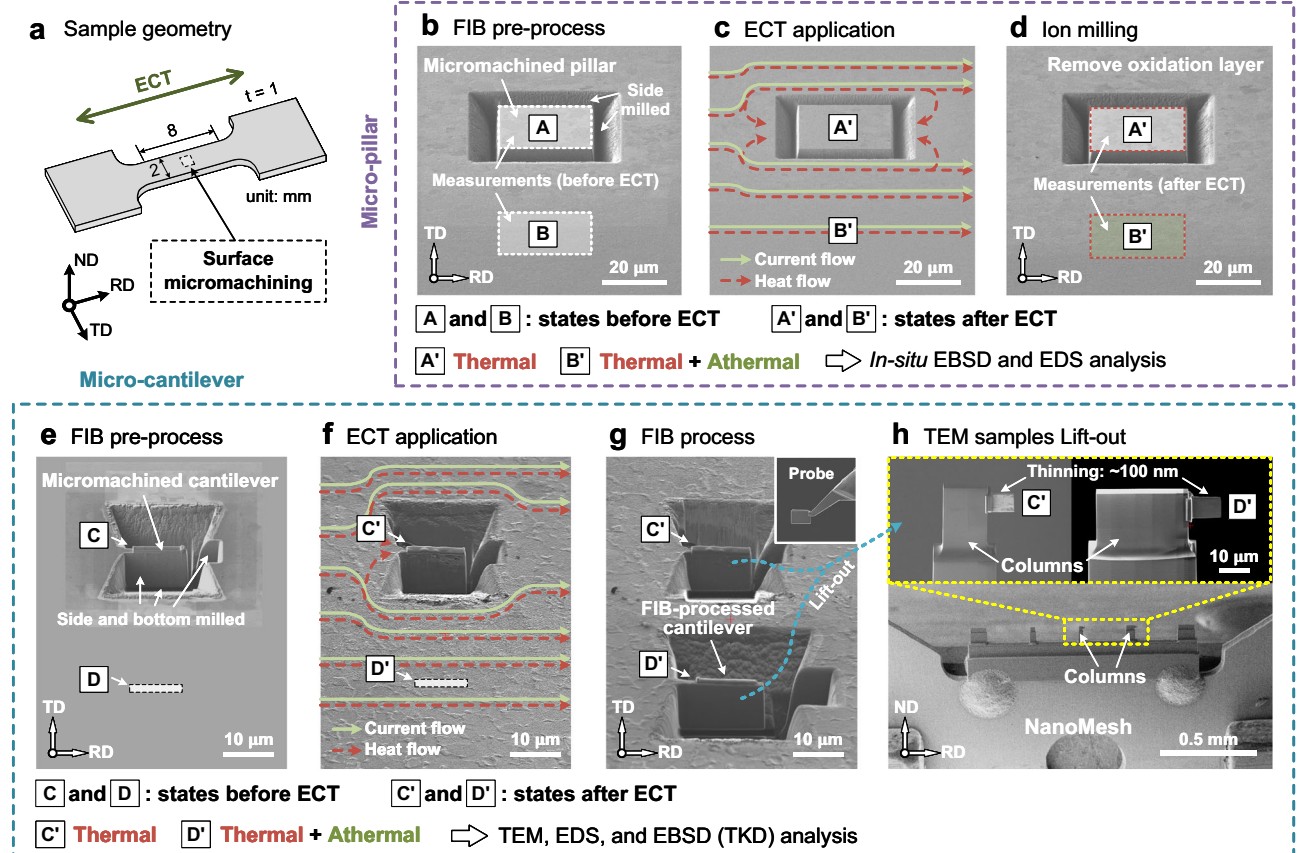

**Fig. 1 | Schematics revealing the thermal and athermal effects using micromachined structures. a** Schematic of the sample geometry. Micromachining is performed on the surface of the central area of the sample, and RD, ND, and TD represent the rolling, normal, and transverse directions, respectively. The sample and current directions are toward the RD direction. **b** Pre-processed micro-pillar using the FIB method, where markers A and B represent the states before the ECT application. **c** Schematic of the current and heat flows near the micromachined pillar and matrix region, marked as A' and B', respectively. **d** Removal of the oxide layer (caused by Joule heating) using ion milling method. **e** Pre-processed micro-cantilever using the FIB method, where markers C and D represent the states before the ECT application. **f** Schematic of the current and heat flows at the micromachined cantilever (C') and matrix region (D'). **g** FIB-processed cantilever in the matrix region. **h** Further processing for TEM observation; the extracted cantilevers are fixed onto the columns of a NanoMesh and thinned down to approximately 100 nm.

at the central area of the sample. Additionally, the schematics of the current cut-off and heat conduction near the micro-cantilever under ECT are illustrated in Fig. 3b, c. A two-dimensional FE model of the cantilever, as provided in Supplementary Note 1, was developed to investigate the effect of the cantilever size on the current and temperature distributions.

We simulated the current density distributions near the cantilever for different $H$ (or $L$) values with a constant $H/L$ rate of 1.5 (Fig. 3d). The relationship between the maximum current density in the central area of the cantilever and the $H$ values (Fig. 3e) demonstrated that the current blocking effect was solely geometry-dependent ($H/L$) and independent of the structure's size. For a constant $H/L = 1.5$, the maximum current value in the central area (Node 1, 7 A mm$^{-2}$) of the cantilever was only ~1% of that in the matrix region. Figure 3f, g show the corresponding temperature distributions and the relationship between the maximum temperature of the cantilever and $H$ values. For a cantilever length ($H$) of ≤30 μm, the difference between the maximum temperature in the central area of the cantilever and matrix was <2%. However, owing to the high costs and technical challenges associated with FIB-assisted fabrication and extraction of cantilever samples for TEM analyses, we adopted a cantilever with dimensions of $H \times L \times T = 15 \times 10 \times 2$ μm$^3$. For this cantilever, the maximum current density in the central area was only 1% of that in the matrix (Fig. 3h), and the temperature difference was only 0.6% (Fig. 3i). This result suggested that the thermal effect on the micromachined cantilever was the same as that on the matrix, whereas the athermal effect was negligible.

## Microstructural variations caused by thermal and athermal effects

Figure 4 shows the in situ EBSD and EDS results obtained before and after the pulsed ECT on the pillar top (thermal effect) and matrix (thermal and athermal effects). Figure 4a, b present the band contrast (BC) maps with the γ phase (in pink) at the pillar top and matrix before and after the pulsed ECT. In addition, we fabricated additional nine micro-pillars, which were characterized using EBSD and EDS. The matrix region was subjected to extensive EBSD and X-ray diffraction (XRD) analyses, as shown in Supplementary Figs. 2–5. The results indicated that the phase transformation between α and γ phases occurred at the pillar top and matrix. The statistical results of 10 micro-pillars showed that the γ phase content in the pillar top increased by an average of 2.9%, whereas it increased by 7.6% (EBSD: 9.7%, XRD: 5.6%) in the matrix region, as shown in Fig. 4e (detailed in Supplementary Figs. 3–4). Therefore, the 2.9% phase transformation at the pillar top was attributed to the thermal effect, whereas the phase transformation, which was different from that in the matrix region (4.7%), was ascribed to the athermal effect. This result indicates that in the phase transformation, the contribution of the athermal effect surpasses that of the thermal effect.

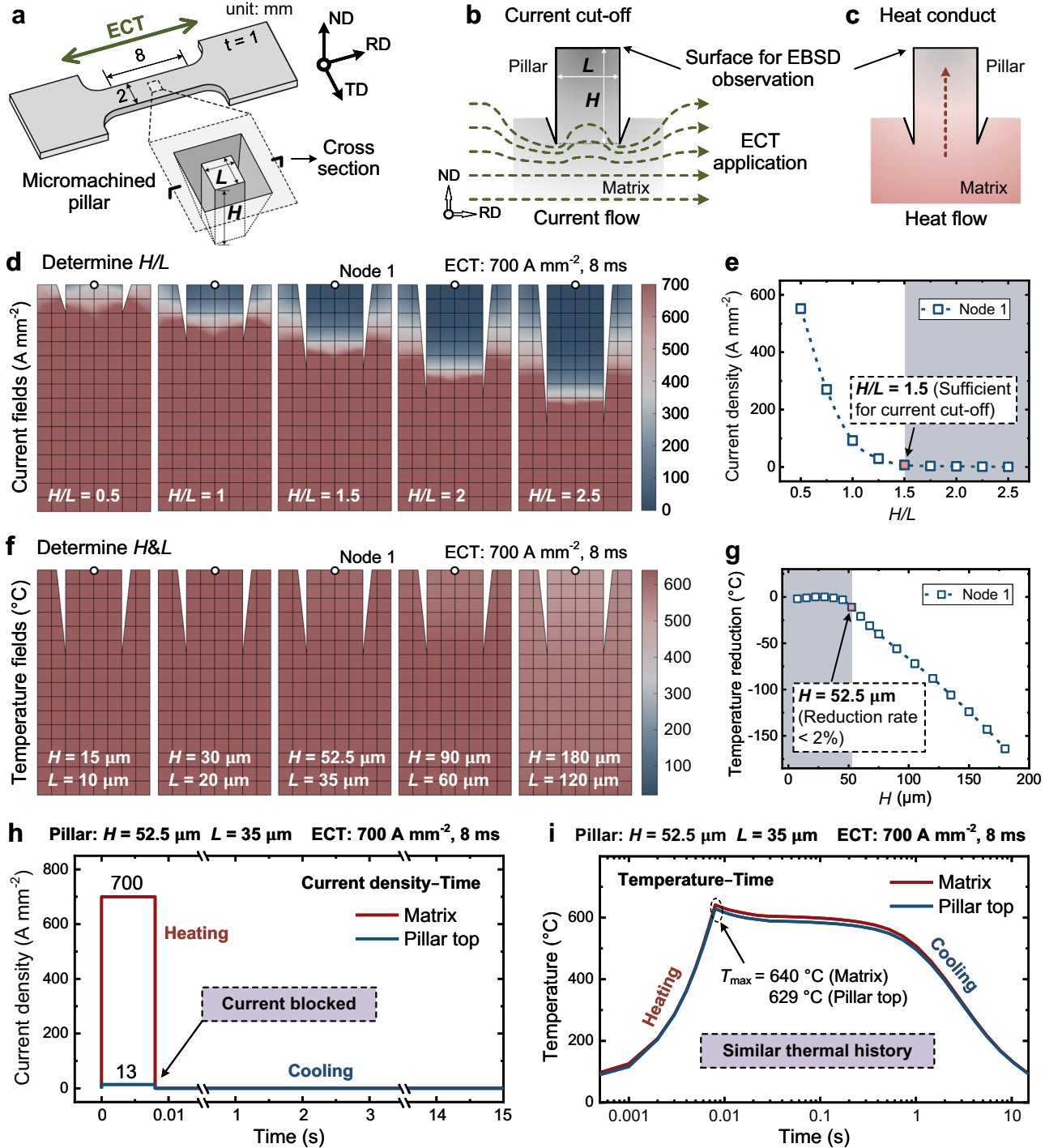

**Fig. 2 | Determining the critical size of the micromachined pillar. a** Schematic of the sample geometry with a micromachined pillar (height: $H$ and length/width: $L$). **b** Current cut-off and **c** Heat conduct from the matrix to the pillar. **d** Current density fields near the pillar of different $H/L$ values under a pulsed ECT at a current density of 700 A mm$^{-2}$ for 8 ms. **e** Plots of the maximum current density at the pillar top with different $H/L$. **f** Temperature fields near the pillar of different $H$ (or $L$) values (constant $H/L = 1.5$). **g** Plots of the maximum temperatures at the pillar top with different $H$ values. **h** Current density–time and **i** temperature–time curves of the matrix region and pillar top of a micromachined pillar with size $H \times L \times L = 52.5 \times 35 \times 35$ µm$^3$ treated at a current density of 700 A mm$^{-2}$ for 8 ms.

The phase transformation conforms to body- and face-centered cubic orientation relationships and increases the lengths of the coherent phase boundaries (CPBs, blue lines in Fig. 4a, b; Supplementary Fig. 6). Similar to the phase transformation results, the changes in the CPBs were also more pronounced in the matrix region than at the pillar top: the length of CPBs per unit area in the matrix region increased from 0.10 to 0.17 µm$^{-1}$ and that at the pillar top increased from 0.12 to 0.15 µm$^{-1}$ (Supplementary Fig. 3).

The distributions of Cr and Ni before and after the pulsed ECT are presented in Fig. 4c, d, and the distributions of Mn and Cu are shown in Supplementary Fig. 5. In the initial microstructure before the ECT (A and B regions), Cr was enriched in the α phase, while Ni, Mn, and Cu were enriched in the γ phase. However, these alloying elements exhibited a mixing phenomenon after the ECT. Cr diffused from the α phase to the γ phase, and conversely, Ni, Mn, and Cu diffused from the γ phase to the α phase. Figure 4f, g present the

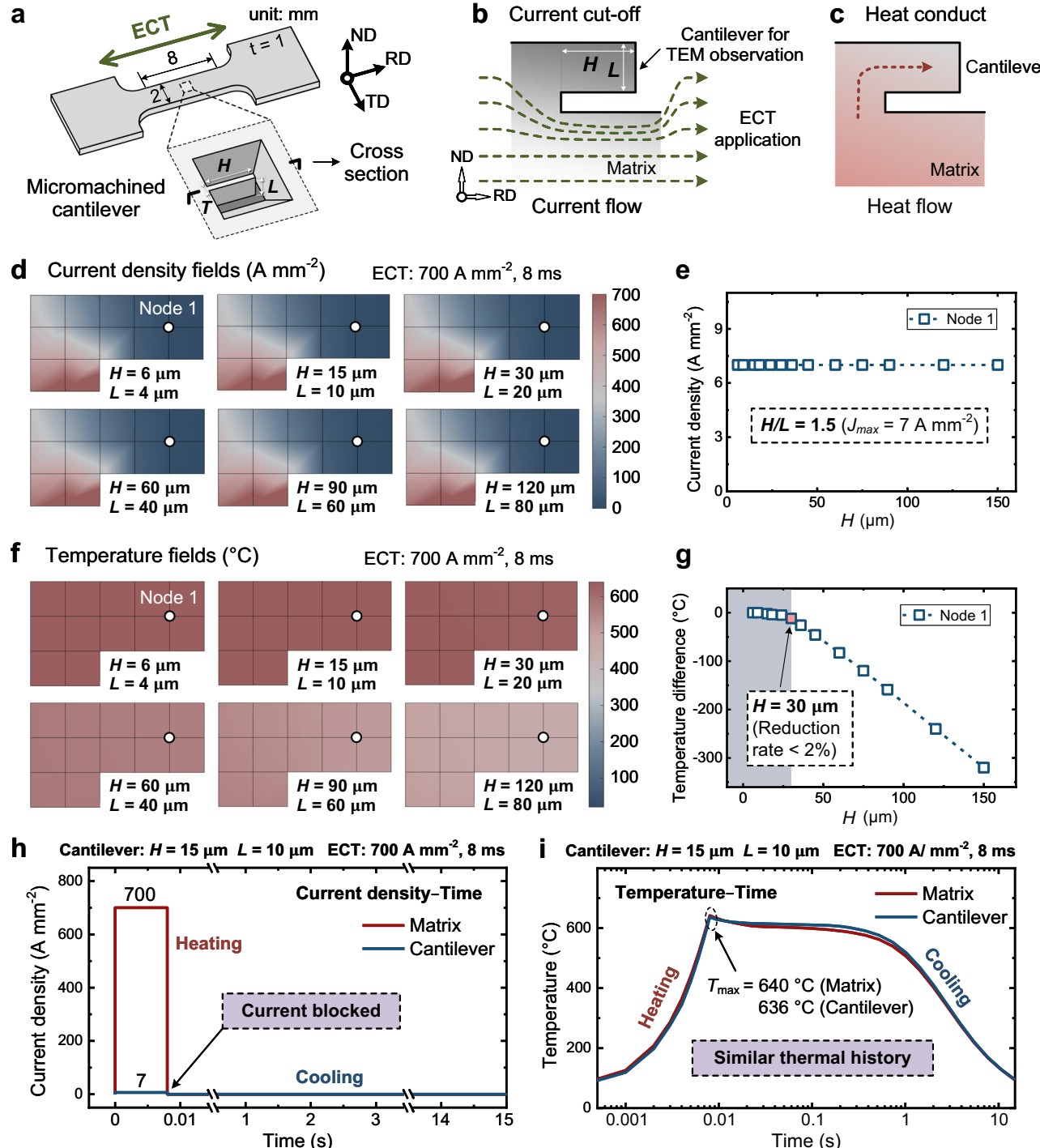

**Fig. 3 | Determining the size of the micromachined cantilever. a** Schematic of the sample geometry with a micromachined cantilever (length: $H$, height: $L$, and thickness: $T$). **b** Current cut-off and **c** heat conduct from the matrix to the cantilever. **d** Current density fields near the cantilever of different $H$ (or $L$) values (constant $H/L = 1.5$) under pulsed ECT at a current density of 700 A mm$^{-2}$ for 8 ms. **e** Plots of the maximum current density at the central area of the cantilever with different $H$ values. **f** Temperature fields near the cantilever with different $H$ (or $L$) values (constant $H/L = 1.5$). **g** Plots of the maximum temperatures at the central area of the cantilever with different $H$ values. **h** Current density– and **i** temperature–time curves at the matrix region and cantilever center of a micromachined cantilever with size $H \times L \times T = 15 \times 10 \times 2$ μm$^3$ at a current density of 700 A mm$^{-2}$ for 8 ms.

changes in the content of Cr within the α phase and those of Ni, Mn, and Cu within the γ phase; these values were obtained from 40 sampling points before and after the ECT (Supplementary Fig. 5). Because the matrix regions show the same thermal history as that of the micromachined structures but experience an additional athermal effect, changes in the elemental content in the micromachined regions may be attributed to the thermal

effect, while differences in the elemental content between micromachined structures and the matrix should be ascribed to the athermal effect. Thus, these results reveal that the thermal and athermal effects decrease the Cr content (α phase) by 0.19% and 0.06%, respectively, as well as decrease the Ni, Mn, and Cu contents (γ phase) by 0.21% and 0.14%, respectively, as shown in Fig. 4f, g.

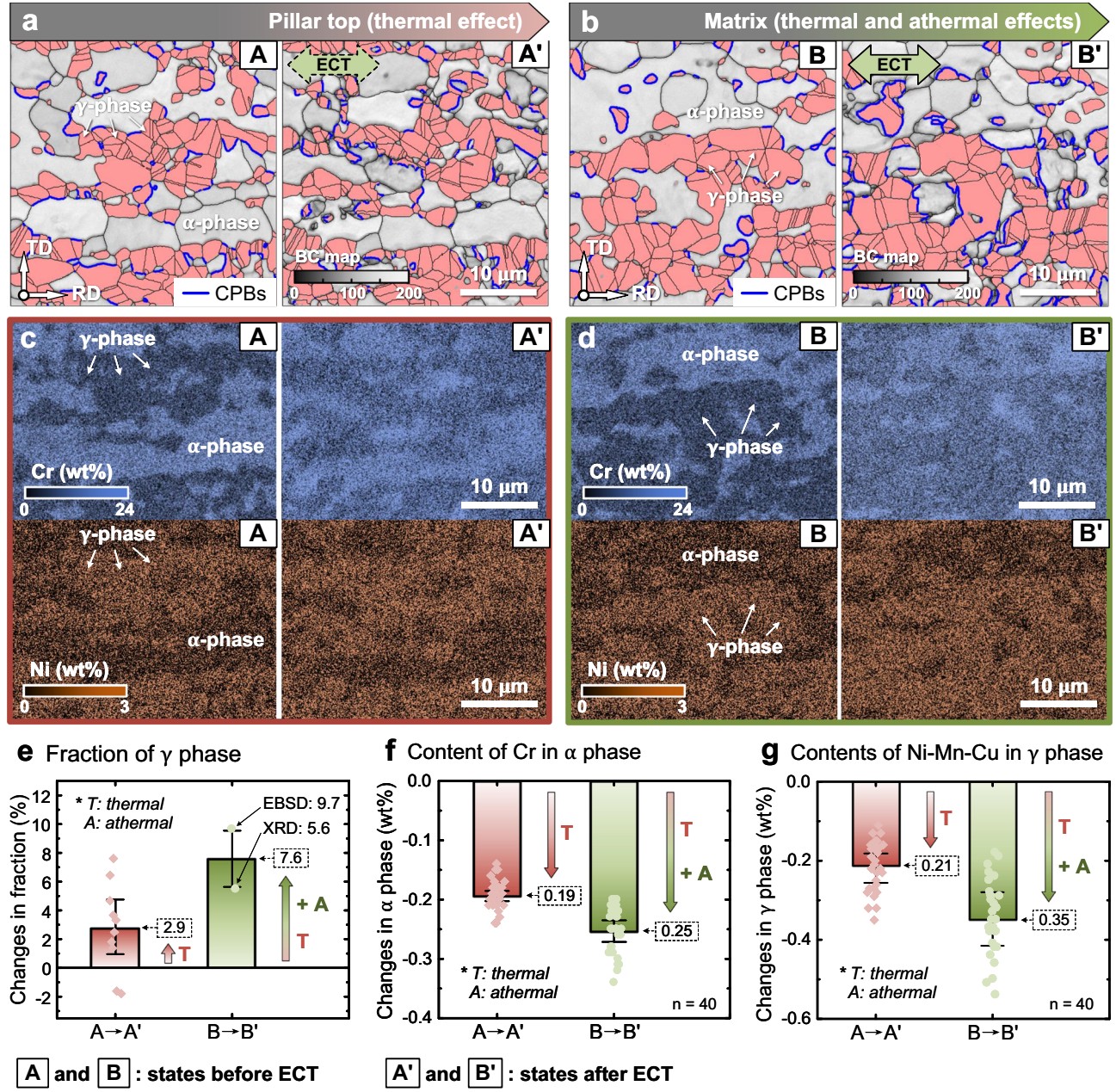

**Fig. 4 | In situ EBSD and EDS analyses on the pillar top and matrix region before and after pulsed ECT. a, b** BC maps with the γ phase (pink) at the pillar top and matrix before and after pulsed ECT, where the CPBs are shown in blue lines. **c, d** Variations of Cr and Ni distributions. **e** Changes in the γ phase content. **f** Content change of Cr in the α phase. **g** Content changes of Ni, Mn, and Cu in the γ phase. Statistical results are mean ± 95% confidence interval (CI).

Figure 5 illustrates the TEM, EDS, and TKD analyses on the micro-cantilever (thermal effect) and matrix (thermal and athermal effects) before and after pulsed ECT. Figure 5a–c presents the scanning transmission electron microscopy (STEM) images of the micro-structure (C/D) before ECT and the micro-cantilever (C′) and matrix (D′) after ECT. The BC maps (γ phase shown in pink and σ phase shown in yellow) in localized regions (Fig. 5a–c) were examined using the TKD method, and the corresponding elemental distributions were assessed using EDS, as illustrated in Fig. 5d–f. The elements tend to mix after ECT. However, several tiny Cr-enriched areas precipitated for the micro-cantilever (C′) were only subjected to the thermal effect, as shown in Fig. 5e, g, which were determined to be the σ phases through elemental quantitative analysis. No precipitation of the σ phase was detected at the matrix region (D′) subjected to both thermal and athermal effects (Fig. 5f).

Additionally, the quantification results of the alloying elements (Cr, Ni, Mn, and Cu) were obtained from 20 sampling points before and after the ECT (Fig. 5h, i). The Cr content within the α phase of the cantilever decreased more than that in the matrix region (Fig. 5h), which may be related to the precipitation of the σ phase. The distribution and diffusion of the γ phase stabilizer elements, such as Ni, Mn, and Cu, were unaffected by the precipitation of the σ phase. A quantitative analysis of the atomic diffusion of Ni, Mn, and Cu in the γ phase was carried out using TEM-EDS. The statistical results showed that the thermal and athermal effects decreased the contents of Ni, Mn, and Cu by 0.18% and 0.24%, respectively, as shown in Fig. 5i, which indicates that the thermal and athermal effects reduced the contents of Ni, Mn, and Cu by 43% and 57%, respectively. Supplementary Table 1 lists the changes in the concentrations of Ni, Mn, and Cu.

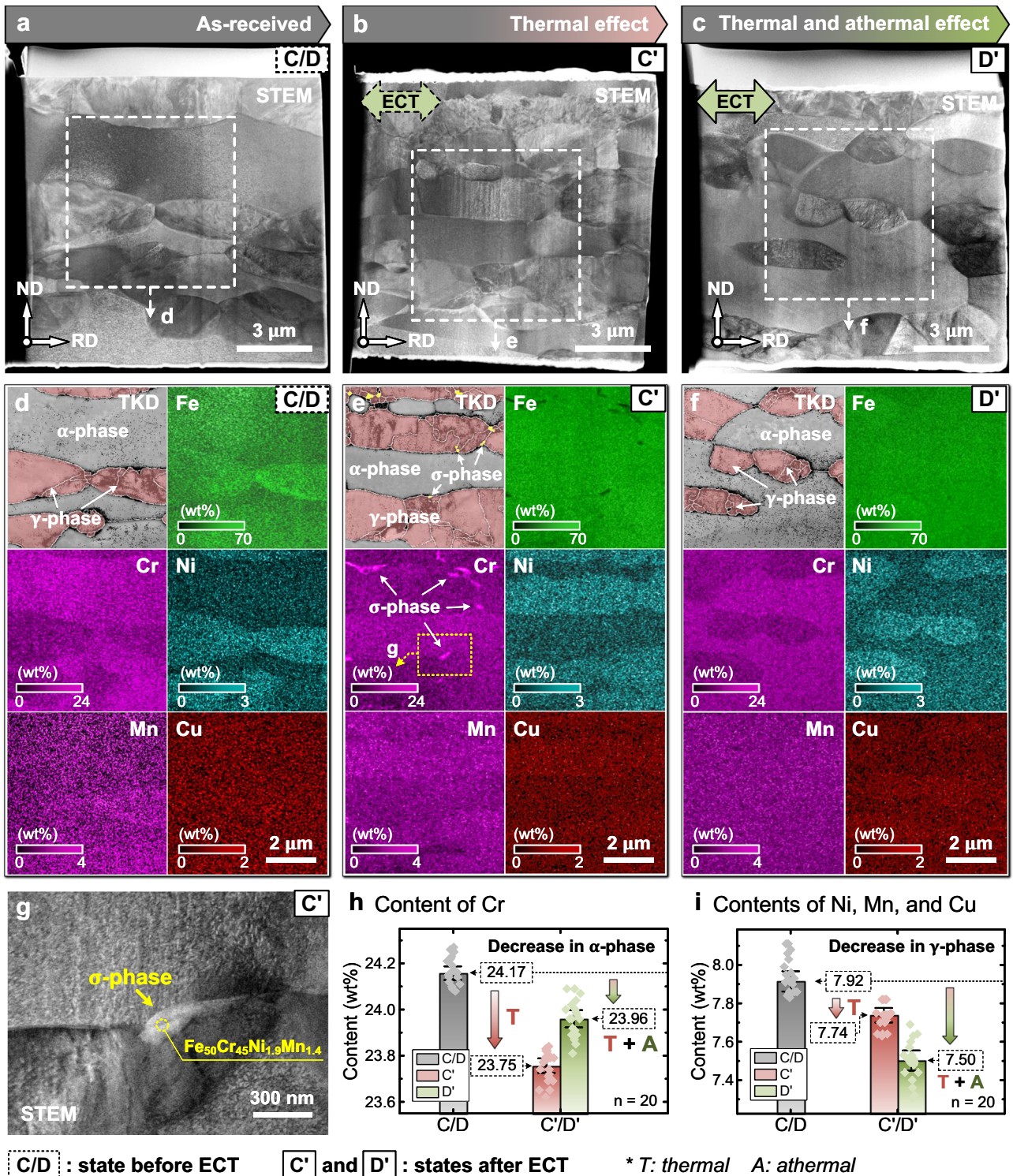

**Fig. 5 | TEM, EDS, and TKD analyses of the micro-cantilever and matrix before and after pulsed ECT. a–c** STEM images of the microstructure (C/D) before ECT, and the micro-cantilever (C') and matrix (D') after ECT. **d–f** TKD analyses (γ phase shown in pink and σ phase in yellow) and elemental distributions of the local areas in (**a–c**). **g** Local STEM image at the area in (**e**), showing the precipitation of σ phase. **h** Content change of Cr in the α phase. **i** Content changes of Ni, Mn, and Cu in the γ phase. Statistical results are mean ± 95% CI.

## Discussion

The diffusion and distribution of the γ-phase stabilizing elements of DSS, such as Ni, Mn, and Cu, used in this study, directly affect the microstructure of the DSS. After applying the pulsed ECT, significant changes occur in the microstructure, including phase transformation and related CPBs and the distribution of the alloying elements, as shown in Figs. 4 and 5. Additionally, we observed the emergence of numerous hemline-like (ring- or crescent-shaped) structures around the γ phase after the ECT in the matrix region, which was subjected to both thermal and athermal effects (B'), as indicated by the yellow arrows in Fig. 6a. These hemline-like structures were small α-grains precipitated ahead of the γ phase. However, these special structures were not observed in the

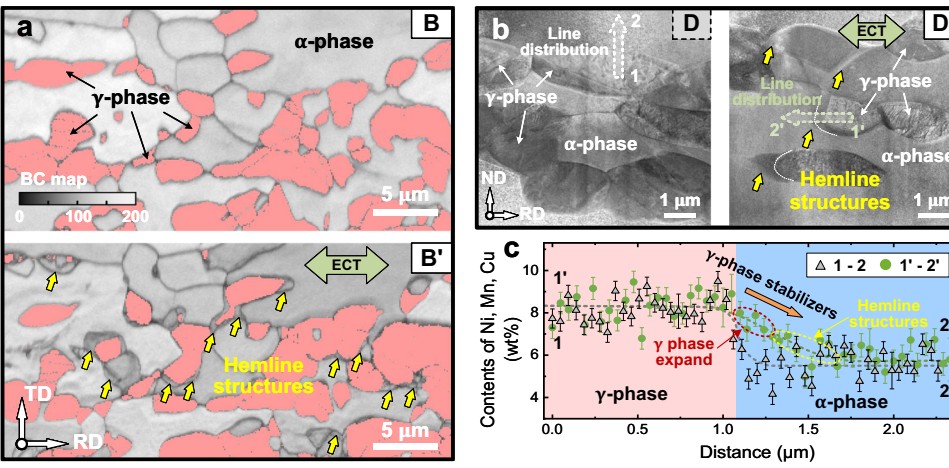

**Fig. 6 | Elemental diffusion and phase transformation caused by pulsed ECT.** **a** BC maps with the γ phase (in pink) of the matrix region (B/B') before and after ECT, where the yellow arrows indicate the hemline structures. **b** STEM images of the cantilevers in the matrix region (D/D') before and after the ECT. **c** Line distributions of the γ phase stabilizers (Ni, Mn, and Cu) along the dashed arrows in (**b**), where the error bars are standard deviations.

micromachined structures, which were subjected only to the thermal effect, and TEM observations also confirmed this phenomenon (Fig. 6b). To investigate this phenomenon, we conducted line analyses of the element distribution (TEM-EDS) from the γ phase to the α phase before and after the ECT, and the corresponding results are indicated by dashed arrows in Fig. 6b. Figure 6c displays the changes in the distributions of the γ-phase stabilizing elements Ni, Mn, and Cu from the γ phase to α phase. Before the ECT, the γ-phase stabilizer content sharply decreased from ~8.2% (γ phase) to ~5.7% (α phase). The steep element distribution slope, observed at the boundary between the γ and α phases, disappeared after the ECT. The diffusion of the γ-phase stabilizer from the γ phase to α phase resulted in the lengthening of the slope. The EBSD and TEM results indicated that the width of this slope (transition range) could increase up to 1 μm. Within this transition range, the γ-phase grains expand with the diffusion of γ-phase stabilizing elements toward the periphery. Moreover, at the forefront of the transition range (on the side contacting the α phase), numerous small α-grains (i.e., the hemline-like structure) precipitate, possibly because the amount of γ-phase stabilizing elements is insufficient to trigger the α-to-γ phase transition (Fig. 6b, c); instead, the precipitation of small α-grains occurs. This diffusion phenomenon, observed at the interface between the α and γ phases under the effect of the electric current, is consistent with the characteristics of electromigration[29]; that is, atoms at the grain boundaries are more prone to migration and diffusion than those within the grains. In addition, we believe that no apparent local heating occurs at the grain boundaries[51] to facilitate the diffusion of elements due to the rapid heat conduction during the ECT. Accordingly, the element diffusion between α and γ phases is quantitatively investigated based on the uniform spatial heat distribution in the following discussion.

In this study, the diffusion of elements under the influence of ECT is associated with the EWF (athermal effect) and compositional gradients (thermal effect). Additionally, other athermal effects apart from the EWF may also contribute to the diffusion of elements. However, due to the lack of experimental and theoretical evidence, these effects will not be discussed in this section. Hence, the theoretical formulas of EWF- and thermal-induced diffusion are expressed as follows[13,29]:

$$J = J_E + J_C = \left( \frac{ND_f}{kT} Z^* e \rho_r j_0 + ND_f \frac{\partial \ln N}{\partial x} \right) \times n_p \qquad (1)$$

where $J$, $J_E$, and $J_C$ are the total, EWF-, and composition gradient-induced atomic fluxes, respectively; $N$, $D_f$, and $k$ represent the atomic density, diffusion coefficient, and Boltzmann's constant, respectively;

$T$ is the temperature in Kelvin; in this study, a constant temperature, $T_{eff}$, was adopted to equate the temperature variations during the ECT for the calculations; $Z^*$, $e$, $\rho_r$, $j_0$, $x$, and $n_p$ represent the effective valency, charge of an electron, electrical resistivity, current density, diffusion distance, and pulse number, respectively. Herein, all types of diffusions are assumed to occur only between adjacent grains; hence, $x = (d_\alpha + d_\gamma)/2$ is assumed, where $d_\alpha$ and $d_\gamma$ are the mean grain sizes of the α and γ phases, respectively.

As the microstructural changes induced by ECT in this study are primarily associated with the diffusion of γ-phase stabilizing elements (Ni, Mn, and Cu), we quantitatively investigated the phenomenon of element diffusion caused by electric current, particularly the EWF. The data for Ni, Mn, and Cu contents obtained through TEM-EDS are presented in Supplementary Table 1. Therefore, the calculated diffusion coefficients for Ni, Mn, and Cu are $1.4 \times 10^{-15}$, $4.7 \times 10^{-16}$, and $3.5 \times 10^{-15} \, \text{m}^2 \, \text{s}^{-1}$, respectively. The necessary parameters are listed in Supplementary Table 2. The obtained diffusion coefficients for the elements align with the previous studies on electric current-assisted diffusion, which typically fall within the range of $10^{-13} - 10^{-16} \, \text{m}^2 \, \text{s}^{-1}$ [18,52–54].

The contributions of Joule heating (thermal effect) and the EWF (athermal effect) to Ni, Mn, and Cu diffusions in this study can be obtained using Eq. (1), and the resulting contribution rates of Joule heating are 54%, 32%, and 39%, respectively, and those of EWF are 46%, 68%, and 61%, respectively. Consequently, the average contributions of Joule heating and the EWF to the diffusion of the γ-phase stabilizing elements are 42% and 58%, respectively, which align with the results shown in Fig. 5i (contributions of 43% and 57%; Joule heating: 0.18% and EWF: 0.24%). Therefore, the calculations suggest the effectiveness of the method employed in this study, which utilizes microfabricated structures to decouple and quantitatively analyze the thermal and athermal effects on microstructural transformation. Furthermore, in this study, the role of the EWF (athermal effect) is more significant than that of Joule heating (thermal effect) on the microstructural modification of the DSS material.

The metal solid-state phase transformations observed in this study are primarily attributed to thermodynamics due to the significant heat changes induced during the ECT (Figs. 2i and 3i). Additionally, because of the DSS material used in this study (with differences in chemical composition between the α-ferrite and γ-austenite phases), the solid-state phase transformation is likely correlated with the diffusion of the elements between the phases[13]. The statistical analysis of the EBSD results obtained from the top surfaces of 10 micro-pillars subjected to the Joule heating indicates that the austenite phase content increases by

an average of 2.9%, as shown in Fig. 4e. Simultaneously, migration and diffusion of the elements occur within the micromachined structures, as shown in Figs. 4f, g and 5h, i. Therefore, the 2.9% increase in the austenite phase is likely related to the thermodynamics and thermomigration of the elements.

Furthermore, we conducted several rapid annealing experiments, in which the samples were immersed in a pre-heated furnace (640 °C) for 1, 2, and 5 min and were then subjected to air cooling. The temperature–time curves of these rapidly annealed samples were estimated using a one-dimensional transient heat conduction model[55]. The results showed that after approximately 1 min of soaking, the sample temperature increased to 635 °C (close to the furnace temperature of 640 °C), as shown in Supplementary Fig. 7. XRD analyses revealed that after 1, 2, and 5 min of rapid annealing, the austenite phase content in the microstructure increased (from 49.5%) to 51.0%, 52.0%, and 52.1%, respectively, with a variation of 1.5%, 2.5%, and 2.6% (Supplementary Fig. 7), respectively. When the annealing time exceeded 2 min, no further increase in the austenite phase content was detected, indicating that it reached a thermal equilibrium state at a temperature of 640 °C. Thus, the pillar top also showed a 2.9% increase in austenite phase content; this result possibly indicated that it reached a similar thermal equilibrium state after 20 times of ECT.

In contrast, in the matrix region (subjected to both Joule heating and the EWF), the austenite phase content increased by approximately 7.6% (based on extensive EBSD and XRD analyses), which was significantly higher than that observed at the top region of the micropillars (2.9%; solely affected by the Joule heating). This result indicates that the EWF plays a dominant role in the phase transformation of DSS materials because of the accelerated migration of elements between the phases, especially that of the austenite-stabilizing elements, due to the combined effects of Joule heating and the EWF (Figs. 4f, g and 5h, i), thereby triggering the austenite transformation. Additionally, the EWF may reduce the activation energy and energy barrier of the solid-state phase transformation[15,56], significantly increasing the austenite phase content. Due to the lack of evidence, whether the athermal effect other than the EWF affects the α-γ phase transition will not be discussed in this study.

The DSS annealing within the temperature range 600 °C to 1000 °C often results in the precipitation of detrimental second phases, such as σ phase, χ phase, nitride, and carbide[57–60]. In this study, the micro-cantilever subjected only to Joule heating (reaching a maximum temperature of approximately 640 °C) exhibited significant precipitation of σ phase (Fig. 5e, g). The Joule heating-induced precipitation of the σ phase (including nucleation and precipitation), which typically contains high levels of Cr and Mo, is attributed to the thermodynamics and elemental diffusion[60], which is random and nondirectional. In this study, the precipitation of the σ phase is primarily associated with Cr. Additionally, previous studies show that a non-isothermal cyclic heat treatment (maximum temperature reaches 600 °C) can accelerate the precipitation of σ phase in DSS materials[61]. In this study, the micromachined DSS structures were also subjected to a similar non-isothermal heat treatment at a maximum temperature of approximately 640 °C for 20 times with an interval of 1 min; the non-isothermal heat-treated micromachined structures showed the precipitation of a large amount of σ phase.

Despite having the same thermal history as that of the microcantilever, the matrix region does not show σ or other secondary phases (Fig. 5f), indicating that the EWF interrupted the nucleation of the σ phase. Recent studies indicate that the ECT has the opposite effect on secondary phases or particles. First, the ECT can dissolve secondary phases or particles, and this phenomenon has been applied in electrically assisted solution heat treatments[62,63]. The ECT can also accelerate the diffusion of alloying elements, thus promoting the nucleation and growth of precipitates, which have been investigated for electrically assisted aging of precipitation-hardened materials[64–66].

Thus, the ECT demonstrates different effects on secondary phases or particles, and the corresponding key factors depend on the material and applied current conditions. Studies on precipitation show that the promotion of nucleation and growth of the secondary phases often occurs under alternating current treatments with low current densities (less than 20 A mm$^{-2}$)[64–66], which activate the diffusion and movement of alloying elements. These features promote elemental aggregation of secondary phases or particles and finally cause the nucleation and growth of second phases. However, when the current density exceeds a certain threshold, the precipitation of the secondary phases or particles is inhibited[67], and these phases or particles even dissolve[62,63] possibly owing to the counteracting effects of the alternating current-induced reciprocating disturbance of the elements and elemental aggregation caused by the thermal equilibrium. Moreover, when the alternating current is switched to a direct current, the secondary phases or particles dissolve, rather than precipitating[68–71] because the concentration gradient of the elements between the secondary phase or particles and the matrix provides a driving force for diffusion (i.e., electromigration), disrupting their thermal equilibrium. Additionally, some in situ observations show that the secondary phases or particles gradually dissolve and peel off from the side facing the electron wind; this result demonstrates the directional characteristics of the ECT[69,70]. Therefore, when a direct current with a current density of up to 700 A mm$^{-2}$ is applied, the DSS material used in this study should exhibit secondary-phase dissolution. Compared with the randomness of the atomic diffusion promoted by the Joule heating, the EWF directionally accelerates atomic diffusion, thus inhibiting the aggregation of Cr and precipitation of the σ phases. Consequently, the ECT method is superior to the conventional heat treatment methods for annealing DSS materials, offering an alternative approach for treating DSS materials at 600–1000 °C without precipitating harmful second phases.

The micromachined structures under the influence of electric current in this study can be simplified into a one-dimensional heat transfer problem as depicted in Fig. 7a, in which on the left side is the matrix (labeled as 1), and on the right side is the interconnected micromachined structure (labeled as 2). The length of the micromachined structure is represented by $h$, signifying the length of the thermal conduction path (TCP). The matrix rapidly heats up with an increase of $\Delta T$ due to Joule heating, while the micromachined structure experiences forced heating due to heat conduction. Thus, the temperature of the micromachined structure always remains lower than that of the matrix. To ensure that the temperature at the end of the micromachined structure does not significantly decrease, the length of the TCP must be limited. Assuming a temperature decrease rate $\eta$, the temperature increase at the end of the micromachined structure is $(1 - \eta)\triangle T$. Consequently, the theoretically calculated critical TCP length is $h = \sqrt{\alpha t/(2 - \eta)}$, where $\alpha$ is the thermal diffusivity of materials and $t$ is the duration of the ECT. Detailed theoretical derivation is shown in Supplementary Note 2.

However, the simplified model overlooks the temperature reduction at position 1 in the matrix due to heat conduction and the heat dissipation in the micromachined structure through heat convection and radiation to the surrounding environment. Therefore, the actual critical TCP length $h_c$ should be smaller than the theoretical value. For instance, in the case of the DSS material used in this study under ECT with a duration of 8 ms, the theoretical critical TCP length $h$ is calculated as 150 μm. However, this value exceeds the FE simulation results, indicating that the TCP lengths of the micro-pillar and micro-cantilever are approximately 50 μm (Figs. 2 and 3). Although Fig. 3 suggests a critical length of 30 μm for the micro-cantilever, considering its connection with the matrix involves bends, the actual TCP length is approximately 50 μm. Thus, accounting for the disparity between the theoretical and actual TCP lengths, we introduce a coefficient $\beta$ and provide a formula for the actual critical TCP length $h_c$ as

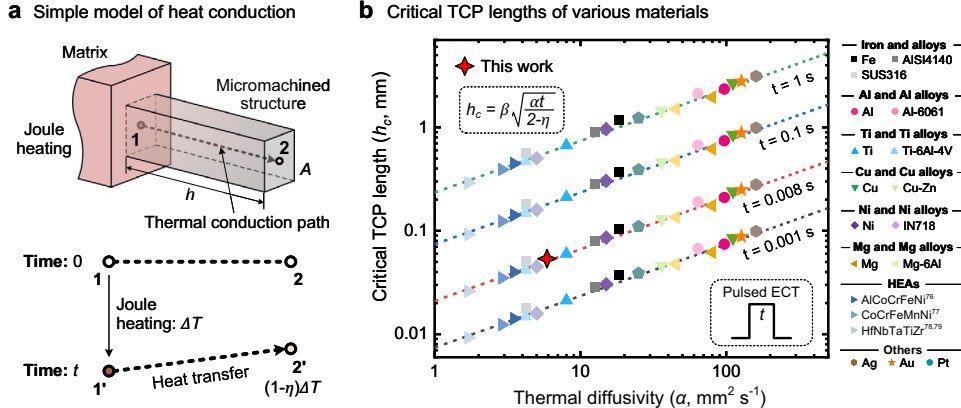

**Fig. 7 | Simple heat conduction model of micromachined structure with critical TCP lengths for various materials. a** Schematic of the simple heat conduction model and heat transfer. **b** Relationship between the critical TCP lengths and the thermal diffusivity of various materials.

expressed below:

$$h_c = \beta \sqrt{\frac{\alpha t}{2 - \eta}} \qquad (2)$$

FE simulations were conducted to determine the critical TCP lengths of various metallic materials, commonly used in engineering structures and the semiconductor industry, under different ECT conditions, and the corresponding results are shown in Fig. 7b. The FE models presented in Supplementary Note 1 were used, and temperature-dependent material properties were employed in the simulation[72–79]. Furthermore, for different materials, the current conditions were set to attain the highest temperature, which was half of the melting point of the materials. The results shown in Fig. 7b indicate that when $\beta = 0.33$, the results obtained using the modified theoretical Eq. (2) align with the simulation results.

Therefore, the micromachined structures processed according to Eq. (2) can ensure that the current is adequately blocked without causing a significant decrease in the temperature compared with that of the matrix. The micromachined structure, whose dimensions are less than the aforementioned critical TCP length, is not affected by the current and exhibits a thermal history similar to that of the matrix. Structures processed based on this critical TCP length can be utilized to distinguish and decouple the influences of the thermal and athermal effects on modified microstructures. Additionally, we can design and adjust the parameters, such as the dimensions of the micromachined structure, maximum temperature, and current conditions, according to Eq. (2) for different experimental purposes. This formula can serve as a reference for quantitatively revealing the contributions of thermal and athermal effects in the microstructural modification of different materials.

## Methods

### Micromachining and experimental procedure

The dumbbell-shaped samples used in this study were prepared by wire cutting from a 1-mm-thick lean DSS sheet, NSSC2120. The measured chemical composition (TEM-based EDS method) was 22.3% Cr, 2.0% Ni, 3.8% Mn, 1.1% Cu, negligible C, and the remaining was Fe; these values were similar to the data provided by the manufacturer. Due to the inclusion of austenite-stabilizing elements (Ni, Mn, and Cu), the DSS comprised approximately 50% α-ferrite and 50% γ-austenite phases. Hence, the α-ferrite phase contained a large amount of Cr and relatively low amounts of austenite-stabilizing elements Ni, Mn, and Cu (24.2% Cr, 1.4% Ni, 3.4% Mn, and 0.9% Cu), whereas the γ-austenite phase contained high levels of Ni, Mn, and Cu (20.4% Cr, 2.5% Ni, 4.2% Mn, and 1.2% Cu). The tensile part of the sample has a length of 8 mm

and a width of 2 mm, aligned with the rolling direction of the sheet, as shown in Supplementary Fig. 1a. Before micromachining, the surface of the samples underwent mechanical and fine chemical polishing. Subsequently, the FIB (ETHOS NX5000, Hitachi) method was employed to fabricate the micro-pillars and micro-cantilevers in the central area of the sample. For the micro-pillars, a gallium ion beam with an accelerating voltage of 30 kV and a current of 45 nA was used to trench around them. Since the height of the pillars was important in this study, the prior examination of the relationship between the designated and actual processing depths (Supplementary Fig. 8) was investigated to ensure accurate trenching. In the selected lean DSS material in this study, the target depth for the micro-pillars was 52.5 μm, as shown in Supplementary Fig. 1b.

Moreover, an ion milling method was employed to remove the oxide layer formed on the sample surface due to the Joule heating caused by the ECT. The IM4000Plus system (Hitachi) was utilized in a flat milling mode, with an acceleration voltage of 6 kV for 5 min, resulting in a removed depth of 167 nm due to the peeling rate of approximately 2 μm h⁻¹. This value is consistent with the observed oxide layer thickness of ~200 nm (Supplementary Fig. 9). EBSD analyses were performed on the micro-pillar top and matrix region before and after ECT. The data was collected under an accelerating voltage of 30 kV and emission current of 18 mA using an EBSD detector (NordlysNano, Oxford Instruments) installed on a scanning electron microscope (SEM, JSM-7200F, JEOL). In addition, the elemental distribution mapping was conducted using an EDS detector (JEOL) installed on the SEM under a 15 kV accelerating voltage and 12 mA emission current. Furthermore, XRD analysis (SmartLab, RIGAKU) was conducted to examine the crystallographic changes in the material before and after the ECT, and the recorded XRD peak intensities were utilized to estimate the changes in the phase content[18].

Moreover, the FIB fabrication of micro-cantilevers included carbon coating, milling, and L-cut, as illustrated in Supplementary Fig. 1c. Further processing of the TEM samples involved lift-out, fixation, and thinning. The TEM sample was fixed onto a column of NanoMesh with a 3-mm diameter for TEM observation using a TEM instrument (TEM, JEM-2100Plus, JEOL) at an acceleration voltage of 200 kV. Meanwhile, the elemental distribution analysis was implemented using an EDS detector (JEOL) mounted on the TEM. Furthermore, the TKD analyses were performed to verify the grain morphology of the FIB-processed TEM samples.

### Electric current application and temperature evaluation

This study employed pulsed ECT due to its constant current value, facilitating subsequent quantitative analysis. In our previous study, we found a significant enhancement in the mechanical performance and

optimization of microstructures after applying 20 pulses of 700 A mm$^{-2}$ for 8 ms on the DSS material used in this study[18]. Thus, we adopted the same condition for 20 pulses, with a one-minute interval between pulses to ensure adequate sample cooling. The electric current was applied using a pulsed power supply (MDA-8000B, MIYACHI). To measure the temperature variations of the sample during pulsed ECT, two infrared thermal sensors were employed: GTL 3 ML-CF4 with a temperature measurement range of 50–400 °C and GTL 2 MH-FF with a range of 380–1600 °C (OPTEX). Under the application of 700 A mm$^{-2}$ for 8 ms, the temperature–time curve at the central area of the sample is shown in Supplementary Fig. 10a, reaching a maximum temperature of ~640 °C. Additionally, a two-dimensional FE model of the sample was established, as illustrated in Supplementary Fig. 10b. A comparison was made between the measured and simulated temperature–time curves at a constant current density of 700 A mm$^{-2}$, with different durations ranging from 6–10 ms (Supplementary Fig. 10a). The results indicated a close alignment between the simulated and measured values. The current density and temperature distributions of the sample are illustrated in Supplementary Fig. 10c, d.

## Data availability

All data required to assess the conclusion of this study are present in this paper and the Supplementary Information. Additional data related to this study may be requested from the authors.

## Code availability

The code used in this study is currently not available for public access. However, interested parties may contact the authors for inquiries regarding the code or potential collaboration.

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

## Acknowledgements

Y.T. acknowledges the financial support from the Japan Science and Technology Agency under the FOREST Program (grant number JPMJFR202B) and the Japan Society for the Promotion of Science under the Grant-in-Aid for Scientific Research (B) (grant number 23K25996). Y.J. acknowledges the support of the Japan Society for the Promotion of Science under the Grant-in-Aid for Scientific Research (S) (grant number 17H06146). S.G. acknowledges the support of the Japan Society for the Promotion of Science under the Grant-in-Aid for Early-Career Scientists (grant number 23K13219) and Research Activity Start-up (grant number 22K20408). This work is partially supported by the Nagoya University Research Fund. The authors are grateful for the assistance provided by the High Voltage Electron Microscope Laboratory, IMaSS, Nagoya University.

## Author contributions

S.G. conceived the idea and designed the experiments. Y.T. guided the research. S.G. conducted the micromachining, microstructural characterization, and data collection. Y.K., X.Y., and C.L. helped with the TEM observation and analysis. S.G. analyzed the data, conducted finite element simulation, and wrote the manuscript. Y.T. and Y.C. contributed to the explanations of the experimental results. All the authors discussed the content of the manuscript. Y.T. and Y.J. supervised the project.

## Competing interests

The authors declare no competing interests.
