## [Peer Review File · Nature Communications]

Micromachined Structures Decoupling Joule Heating and Electron Wind ForceREVIEWER COMMENTS

Reviewer #1 (Remarks to the Author):

The manuscript investigated the thermal and athermal effects in electric current treatments (ECT) by utilizing pre-micromachined structures, which obstruct current flow but maintain a thermal history similar to the matrix. Nowadays, interest in process techniques using electrical energy such as electric current, electrical pulse, and electric field, are growing. Significant efforts are being made to elucidate the mechanism behind microstructural changes induced by electrical current, since understanding the microstructural changes caused by electrical current is particularly crucial for enhancing the efficiency of microstructural control. The authors proposed an innovative and well-designed structure to disentangle the thermal and athermal effects. It was reasonable to determine the shape and size of micro-pillar and micro-cantilever based on FE simulation. By comparing the microstructure before/after ECT, the effect of electric current was discussed. However, some parts are lack of analysis and discussion. Also, it is difficult to verify the discussion because there is missing information.

1. As authors mentioned in manuscript, microstructural modifications caused by ECT are often attributed to thermal and athermal effects. Since the thermal effect is caused by Joule heating, it is clear without the need to consider any other effect. However, there are various research to investigate the athermal effect of electric current. Some studies, mostly early ones, have suggested that electron wind force is the mechanism behind the athermal effect. However, many recent studies claim that the electron wind effect is inadequate to explain the athermal effect. Therefore, it seems inappropriate to equate or confuse the athermal effect with electron wind force (EWF) in the manuscript.

2. In the same context as above, it is recommended that the introduction of the latest research trends on athermal effects should be covered more deeply in the introduction.

3. In Fig.2 and Fig. 3, FE simulation was conducted to determine the critical size of pillar and cantilever. However, there was no information about simulation such as governing equation, boundary condition, initial condition, and mesh information. It should be added in manuscript (method section) or supplementary information.

4. In Fig. 4e, length variation in the CPBs is illustrated. The fractions of a and b phases will be obtained through EBSD measurement. It will be helpful to understand the effect of electric

current on phase transformation by plotting the phase transformation fraction of each phase (alpha- and gamma-phase) before and after ECP. Is there a special reason for comparing only the lengths of CBPs without comparing the phase fractions?

5. The min and max value in color bar in Fig. 4c and d. Fig. 5d-f even have no color bar displayed of each element's content. In order to compare two images through differences in brightness, it is recommended to clearly specify the absolute values of the color bars as they must be confirmed and compared to be the same. Furthermore, the authors discussed the content of Ni, Mn, and Cu in the gamma-phase in Figures 4g and 5i, but no element mapping images of Mn and Cu were provided, except for Ni.

6. Since the phase transformation kinetics vary depending on the contents of Cr, Ni, and C, chemical composition is essential to consider the temperature (640°C) effect during ECT. There is no information about chemical composition, it is hard to discuss the effectiveness of phase transformation at this electrical condition.

7. In cantilever specimen, sigma-phase was observed in thermal effect area. Generally, microstructural changes can be effectively accelerated by ECT, such as annealing, precipitation, phase transformation. If the sigma phase forms between 600-1000°C and the sample's temperature is expected to reach approximately 640°C in this experiment, it can be anticipated that the fraction of the sigma phase would be further accelerated in the thermal+athermal effect area, because it is the thermodynamically stable path. However, microstructure was measured with the opposite trend. However, microstructure was measured with the opposite trend. It is unclear whether this is due to limitations in local area measurements in TEM. If the observed trend is correct (no observation of sigma-phase in thermal+athermal area), discussion supported by scientific evidence should be needed.

8. In Fig. 7b, critical TCP lengths of various material was represented. Did authors conduct the experiment with these various materials? Otherwise, a reference must be displayed for each symbol.

9. TCP was discussed in last part of discussion section. However, the significance or effectiveness of TCP on understanding thermal and athermal effects is unclear.

10. Did the author also conduct tensile testing? I'm asking because the sample geometry picture shows a dog bone-shaped tensile sample in Fig. 1a and Fig. 2a.

11. Many previous research suggested that electric current effect is closely related to defect, such as grain boundary. The lengthening of the slope of gamma-phase stabilizers

after ECT (Fig. 6c) can be discussed considering this point.

12. The portion of decreased content of element in each phase was displayed by thermal and athermal effects separately (Fig. 4e-g, Fig. 5h,i). I wonder what the authors think each portion means considering electric current effect. Also, I wonder the portion of thermal and athermal effects are affected by pillar shape.

Reviewer #2 (Remarks to the Author):

Please see attached file.

Reviewer #3 (Remarks to the Author):

The topic of the paper, utilizing micromachined structures as a way of disentangling the thermal and athermal effects of electric current treatments, is pertinent to the purpose and scope of the journal. The authors' use of pre-micromachined structures is very unique and the method is explained well and backed up by finite element models and experimentation. In terms of the method, this is a well thought out and written paper. The authors also put thought into choosing duplex stainless steel for their validation material so as to look at possible microstructural changes, elemental diffusion, and phase transformations. However, the conclusions that this paper draws on the athermal effect, while possible, are not backed up by statistically significant data.

The following are my comments and revision recommendations that would need to be addressed prior to publication:

(1) As a methods paper with a set of validation experiments, this is a very nice paper. It is recommended that the authors make clear in the results and discussion sections of the duplex stainless steel experiments that they are potential or exemplary results that could be obtained using this method. The quantitative conclusions that are drawn (length of coherent phase boundaries, diffusion of Cr and Ni, etc.) are based on a single sample and cannot be considered statistically significant.

(2) There appear to be error bars on some of the datapoints in Figures 4e,f,g, and 5h,i but it

is not clear how those were obtained. Additionally, the overlap in the error bars, for example in Figure 4g and 5h, show the elemental content % decrease attributed to the thermal and thermal + athermal, respectively, are within the original error of the starting value. While these trends may be correct, additional samples or larger area scans would need to be performed to provide statistical significance and reliability to the conclusions that are drawn.

(3) In lines 261, the authors state they observe the emergence of numerous hemline-like structures. Could the authors elaborate on what they mean by “hemline-like structures”? From the figure, it appears the arrows are pointing to grain precipitates.

(4) In lines 46-47, the authors state that the athermal effect primarily involves electron wind force. There is much in the literature that refutes an electron wind force and while research shows that an athermal effect is occurring, the cause of the athermal effect is still much debated and could likely be caused by different mechanisms depending on the material system of interest. It is recommended the authors make clear that a leading hypothesis is an electron wind effect unless the authors cite specific research for duplex stainless steel.

(5) In the analysis of the thermal effect of the matrix versus the cantilever, the authors state in lines 184-185 that they experience the same thermal effect. This would be true for bulk Joule heating but have the authors considered possible localized joule heating at grain boundaries/dislocations that may occur with electric field versus without and how that may affect the temperature?

(6) In terms of adding statistical significance to the EBSD results for the pillars, the authors may want to consider taking a much larger scan area prior to performing the FIB of the pillar. This would 1) provide analysis of many more grains and 2) show that the FIB work itself doesn't alter the microstructure. Following that, if changes are only seen in the micropillar and nowhere else, the results would be more significant without the burden of many more samples being tested.

Editorial note: Reviewer 2's attached file is below

The manuscript presents an investigation on Micromachined Structures Decoupling Joule Heating and Electron Wind Force. This study proposes an innovative method using pre-micromachined structures that disentangle the athermal effects by hindering current flow. This study may provide a method for revealing the electron wind force of microstructural alterations. The questions raised in the paper are meaningful, but the evidence is insufficient and the subsequent explanations are insufficient. Some of the experimental results in this paper are confusing, and the mechanism of the influence of athermal and thermal effects on precipitation and phase transition is not elaborated in the discussion. In order to improve the manuscript, the following comments should be addressed:

1. There are some grammar errors in the paper.
2. In Fig. 4, there are newly generated and disappeared austenite, and there is no complete correspondence between the phase transformation and the coherent phase boundaries. Can CPBs statistics indicate the degree of phase transition? The meaning of using this indicator is unclear. Can the effects of heat and electricity be determined by simple addition and subtraction of boundary length? Can phase transition be achieved at 640 °C?
3. Is the EDS statistical data in the paper the average of multiple samples. For Table T1, the other elements in both phases decrease before and after electrical treatment except for the Mn element in the α phase, while the total amount of alloy elements should not change. The data here is unreasonable. The Cr element trend in Fig. 4f is significantly different from that in Fig. 5h, although the Cr-rich phase precipitates in the C region, the Cr element reduction (0.5%) in the α phase is still lower than that in the A region with the same thermal history (1.9%).
4. Can the macroscopic temperature measured by the infrared thermal sensor in the experiment represent the temperature changes of the pre-micromachined structures? Will the temperature difference between simulation and experiment in differences in the precipitation of micromachined pillar and micromachined cantilever?
5. How can the statistical results of Ni, Mn, and Cu content in austenite in Figs. 4-5 be displayed in one graph? Is it a simple sum of the mass percentages of three elements?

Their diffusion coefficients should differ.

6. Is electronic wind the only athermal factor that suppresses precipitated phases? The discussion in this section is too simplistic. There are many athermal effects of current, and it should not be simply assumed that the effect of electron wind inhibits the precipitation of precipitates.

7. line 423: previous work is mentioned, but no references are given.

8. In the discussion, there is a lack of detailed statements on the mechanism of the influence of athermal and thermal effects on precipitation and phase transition.

Responses to Reviewers

Title: Micromachined Structures Decoupling Joule Heating and Electron Wind Force

Dear Reviewers,

We are grateful to the reviewers for their valuable suggests and insightful comments on our manuscript. We have revised the manuscript in accordance with the provided feedback. Our point-by-point responses to the comments of the reviewers have been appended below, and the changes made in the revised manuscript have been highlighted in **red color** for ease of identification.

Reviewer #1:

The manuscript investigated the thermal and athermal effects in electric current treatments (ECT) by utilizing pre-micromachined structures, which obstruct current flow but maintain a thermal history similar to the matrix. Nowadays, interest in process techniques using electrical energy such as electric current, electrical pulse, and electric field, are growing. Significant efforts are being made to elucidate the mechanism behind microstructural changes induced by electrical current, since understanding the microstructural changes caused by electrical current is particularly crucial for enhancing the efficiency of microstructural control. The authors proposed an innovative and well-designed structure to disentangle the thermal and athermal effects. It was reasonable to determine the shape and size of micro-pillar and micro-cantilever based on FE simulation. By comparing the microstructure before/after ECT, the effect of electric current was discussed. However, some parts are lack of analysis and discussion. Also, it is difficult to verify the discussion because there is missing information.

1. As authors mentioned in manuscript, microstructural modifications caused by ECT are often attributed to thermal and athermal effects. Since the thermal effect is caused by Joule heating, it is clear without the need to consider any other effect. However, there are various research to investigate the athermal effect of electric current. Some studies, mostly early ones, have suggested that electron wind force is the mechanism behind the athermal effect. However, many recent studies claim that the electron wind effect is inadequate to explain the athermal effect. Therefore, it seems inappropriate to equate or confuse the athermal effect with electron wind force (EWF) in the manuscript.

Response: Thank you for your comments and suggestions. Our intended meaning may not have been fully conveyed in the manuscript. Recent studies on the principles behind

the athermal effect are mainly focused on electroplasticity (current-dislocation interaction), and the underlying principle may be attributed to the electron wind force (EWF), phonon wind force, thermal stress, etc. However, the duplex stainless steel (DSS) material used in this study has no or few dislocations; rather, the microstructure shows α and γ phase compositional differences. Thus, the changes in the microstructure under ECT are more likely related to atomic diffusion, viz. the electromigration phenomenon, associated with the EWF. Hence, the mechanisms of thermal and athermal effects revealed in this study can be specifically attributed to Joule heating and the EWF, respectively. Accordingly, we have made the following changes in the revised manuscript:

Page 4, lines 88–95:

Furthermore, a completely annealed duplex stainless-steel (DSS) material was utilized in this study. The microstructure of the DSS contained approximately 50% α -ferrite and 50% γ -austenite phases with compositional differences in phases that Cr concentrated in the α phase and Ni, Mn, and Cu enriched in the γ phase. Moreover, this DSS contained zero or negligible dislocations. Therefore, the selected material simplifies the athermal effect on the EWF (electromigration-related) without showing effects due to dislocations. Accordingly, the analyzed mechanisms of thermal and athermal effects can be attributed to Joule heating and the EWF, respectively.

2. In the same context as above, it is recommended that the introduction of the latest research trends on athermal effects should be covered more deeply in the introduction.

Response: We appreciate your comments and suggestions. Based on your comments, we have added the analysis and a summary of the literature on the athermal effect in the Introduction as follows:

Page 2, lines 45–56:

The thermal effect primarily involves Joule heating, whereas the athermal effect follows a complex mechanism. Theoretical and experimental evidence pin the electron wind force (EWF) as the origin of the athermal effect in electromigration (atomic diffusion)^{29–31}. However, the principles of electroplasticity (dislocation motion behavior) have not yet been clarified. Initially, Troitskii et al. attributed the motion of electro-induced dislocations to the EWF^{3,4}. Some recent studies have also shown that the EWF plays a decisive role in dislocation motion and reconfiguration^{32,33}. However, the quantified value of the EWF is insufficient to drive dislocation movement, and other causes, such as phonon wind force and thermal gradient or compressive stress, have been proposed^{26,27}. Recent studies suggest that because different materials exhibit different electroplastic phenomena, the mechanisms of the athermal effect underlying this phenomenon might not have a unified explanation^{26–28}.

3. In Fig.2 and Fig. 3, FE simulation was conducted to determine the critical size of pillar and cantilever. However, there was no information about simulation such as governing

equation, boundary condition, initial condition, and mesh information. It should be added in manuscript (method section) or supplementary information.

Response: Thank you for your comments and suggestions. Details of the FE simulation have been added to Supplementary Appendix 1.

Page 6, lines 137–139:

In addition, we established a two-dimensional finite element (FE) model of the pillar, whose detailed governing equations, boundary conditions, and initial conditions are presented in Supplementary Appendix 1.

Page 8, lines 180–182:

A two-dimensional FE model of the cantilever, as provided in Supplementary Appendix 1, was developed to investigate the effect of the cantilever size on the current and temperature distributions.

Appendix 1 in Supplementary Information:

Please refer to the Supplementary Information.

4. In Fig. 4e, length variation in the CPBs is illustrated. The fractions of α and β phases will be obtained through EBSD measurement. It will be helpful to understand the effect of electric current on phase transformation by plotting the phase transformation fraction of each phase (α - and γ -phase) before and after ECP. Is there a special reason for comparing only the lengths of CPBs without comparing the phase fractions?

Response: Thank you for your insightful comments. Due to the limited size of the area (approximately $35\ \mu\text{m} \times 35\ \mu\text{m}$) available for EBSD observation at the top of the micro-pillar, accurately reflecting the microstructural changes based on phase transformation, described in the previous manuscript, is challenging. Additionally, we observed that the phase transformation is a mutual transformation between the α and γ phases. Consequently, solely relying on the changes in the phase content is insufficient to demonstrate the extent of the phase transformation. Furthermore, the transformation between the α and γ phases often follows certain orientation relationships, called coherent phase boundaries (CPBs). Therefore, changes in the CPBs may better reflect the extent of the phase transformation.

Moreover, we have included additional experimental data of nine micro-pillars as well as their EBSD and EDS analysis results in the revised manuscript to address your comments regarding the inadequacy of experimental data, particularly the EBSD data. Both the phase content and CPB changes before and after the ECT have been presented in Fig. 4e in the revised manuscript and in Fig. A3 in Supplementary Information.

Page 10, lines 213–233:

Fig. 4 displays the *in situ* EBSD and EDS results obtained before and after the pulsed

ECT on the pillar top (thermal effect) and matrix (thermal and athermal effects). Figs. 4a and 4b present the band contrast (BC) maps with the γ phase (in pink) at the pillar top and matrix before and after the pulsed ECT. In addition, we fabricated additional nine micro-pillars, which were subsequently characterized using EBSD and EDS techniques. The matrix region was subjected to extensive EBSD and X-ray diffraction (XRD) analyses, as shown in Supplementary Fig. A2–A5. The results indicated that the phase transformation between α and γ phases occurred at the pillar top and matrix. The statistical results of 10 micro-pillars showed that the γ phase content in the pillar top increased by an average of 2.9%, whereas it increased by 7.6% (EBSD: 9.7%, XRD: 5.6%) in the matrix region, as shown in Fig. 4e (detailed in Supplementary Figs. A3–A4). Therefore, the 2.9% phase transformation at the pillar top was attributed to the thermal effect, whereas the phase transformation, which was different from that in the matrix region (4.7%), was ascribed to the athermal effect. This result indicates that in the phase transformation, the contribution of the athermal effect surpasses that of the thermal effect.

The phase transformation conforms to body- and face-centered cubic orientation relationships and increases the lengths of the coherent phase boundaries (CPBs, blue lines in Figs. 4a and 4b; Supplementary Fig. A6). Similar to the phase transformation results, the changes in the CPBs were also more pronounced in the matrix region than at the pillar top: the length of CPBs per unit area in the matrix region increased from 0.10 to 0.17 $\mu\text{m}/\mu\text{m}^2$ and that at the pillar top increased from 0.12 to 0.15 $\mu\text{m}/\mu\text{m}^2$ (Supplementary Fig. A3).

Fig. 4, and Fig. A3 in Supplementary Information:

Please refer to the revised manuscript.

5. The min and max value in color bar in Fig. 4c and d. Fig. 5d-f even have no color bar displayed of each element's content. In order to compare two images through differences in brightness, it is recommended to clearly specify the absolute values of the color bars as they must be confirmed and compared to be the same. Furthermore, the authors discussed the content of Ni, Mn, and Cu in the gamma-phase in Figures 4g and 5i, but no element mapping images of Mn and Cu were provided, except for Ni.

Response: Thank you for your comments and suggestions. In the revised manuscript, we have adjusted our elemental distribution maps according to the measured signal intensity and added color bars. For details, please refer to Figs. 4 and 5. Additionally, we have supplemented the missing elemental distribution maps for Mn and Cu (presented in Supplementary Fig. A5 of SEM-EDS and in Figs. 5d–f of TEM-EDS).

Figs. 4 and 5, and Fig. A5 in Supplementary Information:

Please refer to the revised manuscript.

6. Since the phase transformation kinetics vary depending on the contents of Cr, Ni, and C, chemical composition is essential to consider the temperature (640°C) effect during ECT.

There is no information about chemical composition, it is hard to discuss the effectiveness of phase transformation at this electrical condition.

Response: Thank you for your comments and suggestions. We have incorporated the elemental contents (Cr, Ni, Mn, and Cu) of the as-received material in the Method section in the revised manuscript. Additionally, the changes in the elemental contents after the ECT have been indicated in Figs. 4 and 5 as well as in Supplementary Fig. A5 and Table T1. Furthermore, the C content is negligible, and even the manufacturer has not listed its value; therefore, its impact on the microstructural modification is not considered in this study. The following changes have been made in the revised manuscript:

Pages 21–22, lines 503–511:

The dumbbell-shaped samples used in this study were prepared by wire cutting from a 1-mm-thick lean DSS sheet, NSSC2120. The measured chemical composition (TEM-based EDS method) was 22.3% Cr, 2.0% Ni, 3.8% Mn, 1.1% Cu, negligible C, and the remaining was Fe; these values were similar to the data provided by the manufacturer. Due to the inclusion of austenite-stabilizing elements (Ni, Mn, and Cu), the DSS comprised approximately 50% α -ferrite and 50% γ -austenite phases. Hence, the α -ferrite phase contained a large amount of Cr and relatively low amounts of austenite-stabilizing elements Ni, Mn, and Cu (24.2% Cr, 1.4% Ni, 3.4% Mn, and 0.9% Cu), whereas the γ -austenite phase contained high levels of Ni, Mn, and Cu (20.4% Cr, 2.5% Ni, 4.2% Mn, and 1.2% Cu).

Figs. 4 and 5, and Fig. A5 and Table T1 in Supplementary Information:

Please refer to the revised manuscript.

7. In cantilever specimen, sigma-phase was observed in thermal effect area. Generally, microstructural changes can be effectively accelerated by ECT, such as annealing, precipitation, phase transformation. If the sigma phase forms between 600-1000°C and the sample's temperature is expected to reach approximately 640°C in this experiment, it can be anticipated that the fraction of the sigma phase would be further accelerated in the thermal+athermal effect area, because it is the thermodynamically stable path. However, microstructure was measured with the opposite trend. However, microstructure was measured with the opposite trend. It is unclear whether this is due to limitations in local area measurements in TEM. If the observed trend is correct (no observation of sigma-phase in thermal+athermal area), discussion supported by scientific evidence should be needed.

Response: Thank you for your valuable comments and suggestions. Indeed, as you mentioned, if the σ phase is stable within the temperature range of 600–1000 °C, any stimulus, such as heat, would cause σ phase precipitation along this pathway in the material. The ability of the ECT (thermal + athermal effects) to cause precipitation of secondary phases or particles more effectively compared to the thermal effect depends on

the material and current conditions. Recent research indicates that the ECT exhibits the opposite effect: it can either promote the precipitation of secondary phases or particles, or accelerate their dissolution. Some studies show that an alternating current with a low current density (less than 20 A/mm²) tends to promote precipitation, whereas a large current density and/or direct current (DC) promotes dissolution. Therefore, the 700 A/mm² pulsed DC applied in this study tended to suppress the precipitation of secondary phases or particles. As a result, no σ phase precipitation was observed in the matrix region of the samples. The following changes have been made in the revised manuscript:

Pages 18–19, lines 416–446:

Despite having the same thermal history as that of the micro-cantilever, the matrix region does not show σ or other secondary phases (Fig. 5f), indicating that the EWF interrupted the nucleation of the σ phase. Recent studies indicate that the ECT has the opposite effect on secondary phases or particles. First, the ECT can dissolve secondary phases or particles, and this phenomenon has been applied in electrically assisted solution heat treatments^{63,64}. The ECT can also accelerate the diffusion of alloying elements, thus promoting the nucleation and growth of precipitates, which have now been studied for electrically assisted aging of precipitation-hardened materials^{65–67}. Thus, the ECT demonstrates entirely different effects on secondary phases or particles, and the corresponding key factors depend on the material and applied current conditions. Studies on precipitation show that the promotion of nucleation and growth of the secondary phases often occurs under alternating current treatments with low current densities (less than 20 A/mm²)^{65–67}, which activate the diffusion and movement of alloying elements. These features promote elemental aggregation of secondary phases or particles and finally cause the nucleation and growth of second phases. However, when the current density exceeds a certain threshold, the precipitation of the secondary phases or particles is inhibited⁶⁸, and these phases or particles even dissolve^{63,64} possibly owing to the counteracting effects of the alternating current-induced reciprocating disturbance of the elements and elemental aggregation caused by the thermal equilibrium. Moreover, when the alternating current is switched to a direct current, the secondary phases or particles dissolve, rather than precipitating^{69–72} because the concentration gradient of the elements between the secondary phase or particles and the matrix provides a driving force for diffusion (i.e., electromigration), which disrupts their thermal equilibrium. Additionally, some *In situ* observations show that the secondary phases or particles gradually dissolve and peel off from the side facing the electron wind; this result demonstrates the directional characteristics of the ECT^{70,71}. Therefore, when a direct current with a current density of up to 700 A/mm² is applied, the DSS material used in this study should exhibit secondary-phase dissolution. Unlike the randomness of the atomic diffusion promoted by the Joule heating, the EWF directionally accelerates atomic diffusion, thus inhibiting the aggregation of Cr and precipitation of the σ phases. Consequently, the ECT method is superior to the conventional heat treatment methods for annealing DSS materials, offering an alternative approach for treating DSS materials at 600–1000 °C without precipitating harmful second phases.

8. In Fig. 7b, critical TCP lengths of various material was represented. Did authors conduct the experiment with these various materials? Otherwise, a reference must be displayed for each symbol.

Response: Thank you for your helpful comments. We have not conducted experiments to determine the TCP lengths listed for all the materials in Fig. 7b; instead, the results were obtained using finite element simulation methods. The temperature-dependent physical parameters of common metals, other than high entropy alloys, were adopted from material handbooks and free online databases (mentioned in Line 480), and those of high entropy alloys were adopted from specific literatures, which have been cited in Fig. 7b and in the revised manuscript.

Fig. 7:

Please refer to the revised manuscript.

9. TCP was discussed in last part of discussion section. However, the significance or effectiveness of TCP on understanding thermal and athermal effects is unclear.

Response: Thank you for your helpful comment. The significance of the critical TCP length and its potential application scenarios have been supported with additional explanations in the final part of the Discussion section as indicated below:

Page 21, lines 484–494:

Therefore, the micromachined structures processed according to equation (2) can ensure that the current is adequately blocked without causing a significant decrease in the temperature compared with that of the matrix. The micromachined structure, whose dimensions are less than the aforementioned critical TCP length, is not affected by the current and exhibits a thermal history similar to that of the matrix. Structures processed based on this critical TCP length can be used to distinguish and decouple the influences of the thermal and athermal effects on modified microstructures. Additionally, we can design and adjust the parameters, such as the dimensions of the micromachined structure, maximum temperature, and current conditions, according to equation (2) for different experimental purposes. This critical formula can serve as a reference for quantitatively revealing the contributions of thermal and athermal effects in the microstructural modification of different materials.

10. Did the author also conduct tensile testing? I'm asking because the sample geometry picture shows a dog bone-shaped tensile sample in Fig. 1a and Fig. 2a.

Response: Thank you for your comments. We have conducted tensile tests, and the results have already been published in our previous paper. These previous study results showed that the ECT could significantly improve the material's ductility through the enhanced TRIP effect. Preliminary investigations suggested that this ductility improvement might be attributed to the diffusion of austenite stabilizing elements and

increased contents of the γ -austenite phase. However, gaining further deep insights into this mechanism, especially in terms of how the thermal and athermal effects affect the microstructure, is necessary. Thus, this present study was focused on solving this problem using a novel method based on pre-micromachined structures. Our previous report has been cited in the revised manuscript as indicated below:

Page 23, lines 548–550:

In our previous study, we found a significant enhancement in the mechanical performance and optimization of microstructures after applying 20 pulses of 700 A/mm² for 8 ms on the DSS material used in this study¹⁸.

18. Gu, S. *et al.* Realizing strength–ductility synergy in a lean duplex stainless steel through enhanced TRIP effect via pulsed electric current treatment. *Mater. Sci. Eng. A* **883**, 145534 (2023).
[10.1016/j.msea.2023.145534](https://doi.org/10.1016/j.msea.2023.145534).

11. Many previous research suggested that electric current effect is closely related to defect, such as grain boundary. The lengthening of the slope of gamma-phase stabilizers after ECT (Fig. 6c) can be discussed considering this point.

Response: We appreciate your insightful comments and suggestions. Accordingly, we have added relevant information in the revised manuscript as indicated below:

Pages 14–15, lines 307–321:

The EBSD and TEM results indicated that the width of this slope (transition range) could increase up to 1 μm . Within this transition range, the γ -phase grains expand with the diffusion of γ -phase stabilizing elements toward the periphery. Moreover, at the forefront of the transition range (on the side contacting the α phase), numerous small α -grains (i.e., the hemline-like structure) precipitate, possibly because the amount of γ -phase stabilizing elements is insufficient to trigger the α -to- γ phase transition (Figs. 6b and 6c); instead, the precipitation of small α -grains occurs. This diffusion phenomenon, observed at the interface between the α and γ phases under the effect of the electric current, is consistent with the characteristics of electromigration²⁹; that is, atoms at the grain boundaries are more prone to migration and diffusion than those within the grains. In addition, we believe that no obvious local heating occurs at the grain boundaries⁵² to facilitate the diffusion of elements owing to the rapid heat conduction during the ECT. Accordingly, the element diffusion between α and γ phases is quantitatively investigated based on the uniform spatial heat distribution in the following discussion.

12. The portion of decreased content of element in each phase was displayed by thermal and athermal effects separately (Fig. 4e-g, Fig. 5h,i). I wonder what the authors think each portion means considering electric current effect. Also, I wonder the portion of thermal and athermal effects are affected by pillar shape.

Response: Thank you for your comments. We believe that the changes in the phase

fractions and elemental contents in the micromachined structures should be attributed solely to the thermal effect, whereas the differences in the phase fractions and elemental contents between the micromachined structures and the matrix region should be ascribed to the athermal effect. This is because our simulation results show that the matrix region exhibits the same thermal history as that of the micromachined structures as well as exhibits the athermal effect. Therefore, we consider that these differences in the microstructure changes can be used to indicate the thermal and athermal effects.

Additionally, we believe that the shape of the pillars, cantilevers, or other structures does not influence the elemental diffusion. The TCP is anticipated to be the more important parameter for the micromachined structures, as shown in Fig. 7a; the TCP directly influences the electrical and thermal states of the micromachined structures subjected to ECT. Hence, we have provided a detailed analysis in the last part of the Discussion section, as shown below:

Page 21, lines 484–494:

Therefore, the micromachined structures processed according to equation (2) can ensure that the current is adequately blocked without causing a significant decrease in the temperature compared with that of the matrix. The micromachined structure, whose dimensions are less than the aforementioned critical TCP length, is not affected by the current and exhibits a thermal history similar to that of the matrix. Structures processed based on this critical TCP length can be used to distinguish and decouple the influences of the thermal and athermal effects on modified microstructures. Additionally, we can design and adjust the parameters, such as the dimensions of the micromachined structure, maximum temperature, and current conditions, according to equation (2) for different experimental purposes. This critical formula can serve as a reference for quantitatively revealing the contributions of thermal and athermal effects in the microstructural modification of different materials.

Reviewer #2:

The manuscript presents an investigation on Micromachined Structures Decoupling Joule Heating and Electron Wind Force. This study proposes an innovative method using pre-micromachined structures that disentangle the athermal effects by hindering current flow. This study may provide a method for revealing the electron wind force of microstructural alterations. The questions raised in the paper are meaningful, but the evidence is insufficient and the subsequent explanations are insufficient. Some of the experimental results in this paper are confusing, and the mechanism of the influence of athermal and thermal effects on precipitation and phase transition is not elaborated in the discussion. In order to improve the manuscript, the following comments should be addressed:

1. There are some grammar errors in the paper.

Response: Thank you for your comments. We employed a professional English editing service (Editage) to proofread our manuscript. Additionally, we carefully reviewed the revised manuscript.

2. In Fig. 4, there are newly generated and disappeared austenite, and there is no complete correspondence between the phase transformation and the coherent phase boundaries. Can CPBs statistics indicate the degree of phase transition? The meaning of using this indicator is unclear. Can the effects of heat and electricity be determined by simple addition and subtraction of boundary length? Can phase transition be achieved at 640 °C?

Response: Thank you for your insights and suggestions. Indeed, as mentioned by you, the transitions between the α and γ phases before and after the electric current application are mutual, and the boundaries between the phases are not always CPBs. Owing to limitations in the observation area, accurately representing the effects of the electric current through changes in the phase content is challenging. However, we found that mutual transformations between the α and γ phases often follow certain orientation relationships and then form CPBs. These orientation relationships between the phase transitions have been extensively validated by numerous studies^{R1-R4}. Therefore, the changes in the CPBs may better reflect the extent of phase transformation between the α and γ phases.

However, in the revised manuscript, we have included additional experimental data of nine micro-pillars and their EBSD analysis results obtained before and after the ECT to present both the phase fraction and CPB changes.

Furthermore, (regarding the question of whether 640 °C can induce phase transitions), we conducted several rapid annealing experiments, in which the samples were immersed in a pre-heated furnace (640 °C) for 1, 2, and 5 min. The simulation results show that after 1-min soaking, the sample temperature was close to 640 °C. Additionally, the changes in phase content were detected using the XRD method. The results showed that the γ phase content increased by 1.5%, 2.5%, and 2.6% after soaking for 1, 2, and 5 min,

respectively, indicating that heat treatment at 640 °C could induce phase transitions. The corresponding modifications in the revised manuscript are as follows:

Page 10, lines 213–233:

Fig. 4 displays the *in situ* EBSD and EDS results obtained before and after the pulsed ECT on the pillar top (thermal effect) and matrix (thermal and athermal effects). Figs. 4a and 4b present the band contrast (BC) maps with the γ phase (in pink) at the pillar top and matrix before and after the pulsed ECT. In addition, we fabricated additional nine micro-pillars, which were subsequently characterized using EBSD and EDS techniques. The matrix region was subjected to extensive EBSD and X-ray diffraction (XRD) analyses, as shown in Supplementary Fig. A2–A5. The results indicated that the phase transformation between α and γ phases occurred at the pillar top and matrix. The statistical results of 10 micro-pillars showed that the γ phase content in the pillar top increased by an average of 2.9%, whereas it increased by 7.6% (EBSD: 9.7%, XRD: 5.6%) in the matrix region, as shown in Fig. 4e (detailed in Supplementary Figs. A3–A4). Therefore, the 2.9% phase transformation at the pillar top was attributed to the thermal effect, whereas the phase transformation, which was different from that in the matrix region (4.7%), was ascribed to the athermal effect. This result indicates that in the phase transformation, the contribution of the athermal effect surpasses that of the thermal effect.

The phase transformation conforms to body- and face-centered cubic orientation relationships and increases the lengths of the coherent phase boundaries (CPBs, blue lines in Figs. 4a and 4b; Supplementary Fig. A6). Similar to the phase transformation results, the changes in the CPBs were also more pronounced in the matrix region than at the pillar top: the length of CPBs per unit area in the matrix region increased from 0.10 to 0.17 $\mu\text{m}/\mu\text{m}^2$ and that at the pillar top increased from 0.12 to 0.15 $\mu\text{m}/\mu\text{m}^2$ (Supplementary Fig. A3).

Page 17, lines 377–389:

Furthermore, we conducted several rapid annealing experiments, in which the samples were immersed in a pre-heated furnace (640 °C) for 1, 2, and 5 min and were then subjected to air cooling. The temperature–time curves of these rapidly annealed samples were estimated using a one-dimensional transient heat conduction model⁵⁶. The results showed that after approximately 1 min of soaking, the sample temperature increased to 635 °C (close to the furnace temperature of 640 °C), as shown in Supplementary Fig. A7. XRD analyses revealed that after 1, 2, and 5 min of rapid annealing, the austenite phase content in the microstructure increased (from 49.5%) to 51.0%, 52.0%, and 52.1%, respectively, with a variation of 1.5%, 2.5%, and 2.6% (Supplementary Fig. A7), respectively. When the annealing time exceeded 2 min, no further increase in the austenite phase content was detected, indicating that it reached a thermal equilibrium state at a temperature of 640 °C. Thus, the pillar top also showed a 2.9% increase in austenite phase content; this result possibly indicated that it reached a similar thermal equilibrium state after 20 times of ECT.

Fig. 4, and Fig. A2–A7 in Supplementary Information:

Please refer to the revised manuscript.

[R1] E.F. Monlevade, I.G.S. Falleiros, Orientation relationships associated with austenite formation from ferrite in a coarse-grained duplex stainless steel, *Metall. Mater. Trans. A* **37** (2006) 939–949, <https://doi.org/10.1007/s11661-006-0067-1>.

[R2] X. Fang, W. Yin, C. Qin, W. Wang, K.H. Lo, C.H. Shek, The interface character distribution of cold-rolled and annealed duplex stainless steel, *Mater. Char.* **118** (2016) 397–404, <https://doi.org/10.1016/j.matchar.2016.06.017>.

[R3] N. Haghdadi, P. Cizek, P.D. Hodgson, V. Tari, G.S. Rohrer, H. Beladi, Effect of ferrite-to-austenite phase transformation path on the interface crystallographic character distributions in a duplex stainless steel, *Acta Mater.* **145** (2018) 196–209, <https://doi.org/10.1016/j.actamat.2017.11.057>.

[R4] N. Haghdadi, P. Cizek, P.D. Hodgson, V. Tari, G.S. Rohrer, H. Beladi, Fiveparameter crystallographic characteristics of the interfaces formed during ferrite to austenite transformation in a duplex stainless steel, *Phil. Mag.* **98** (2018) 1284–1306, <https://doi.org/10.1080/14786435.2018.1434321>.

3. Is the EDS statistical data in the paper the average of multiple samples. For Table T1, the other elements in both phases decrease before and after electrical treatment except for the Mn element in the α phase, while the total amount of alloy elements should not change. The data here is unreasonable. The Cr element trend in Fig. 4f is significantly different from that in Fig. 5h, although the Cr-rich phase precipitates in the C region, the Cr element reduction (0.5%) in the α phase is still lower than that in the A region with the same thermal history (1.9%).

Response: Thank you for your insightful comments. In the previous manuscript, we had utilized the quantitative EDS analysis of the elemental content in 3–6 local regions in one sample. In the revised manuscript, we have included data of additional nine micro-pillars, whose top surfaces were subjected to EDS analyses before and after the ECT. We increased the sampling number to 40 for the EDS analysis. Moreover, for the TEM-EDS elemental analysis (Fig. 5), the resolution was sufficiently high, and thus, we did not add extra samples, but collected data from more sampling points (20). The corresponding results are presented in Figs. 4 and 5 in the revised manuscript:

The unreasonable trend of the alloying elements Ni, Mn, and Cu in the α and γ phases, shown in Supplementary Table T1, was due to the insufficient number of sampling points. We have increased the number of sampling points significantly and presented the statistical results in Supplementary Table 1.

Furthermore, the excessive decrease in the Cr content visible in Fig. 4f (1.9%) compared to that in Fig. 5h (0.5%) was influenced by the inaccurate SEM-EDS-based evaluation of the light elements C, N, and O. Due to the low contents of C, N, and O in the materials used in this study, we have omitted them from the quantitative assessments mentioned in the revised manuscript. The obtained results are in good agreement with the elemental diffusion trend observed in the TEM-EDS analysis. The following modifications have

been made in the revised manuscript:

Figs. 4–5, and Table T1 in Supplementary Information:

Please refer to the revised manuscript.

4. Can the macroscopic temperature measured by the infrared thermal sensor in the experiment represent the temperature changes of the pre-micromachined structures? Will the temperature difference between simulation and experiment in differences in the precipitation of micromachined pillar and micromachined cantilever?

Response: Thank you for your comments. First, our macroscopic finite element simulation analysis (Supplementary Fig. A10) results align well with those of the temperature measurements conducted using the thermal sensors. This indicates that the parameters and boundary conditions chosen for the simulation are consistent with actual conditions. Additionally, we believe that using these parameters to simulate micro-pillars and micro-cantilevers would provide reasonable results, although we cannot provide the measured changes in the temperature of the micromachined structures owing to the technical challenges associated with such measurements. However, based on our simulation results, we consider the simulations on micromachined structures to be reasonable. When the dimensions of the micromachined structure are below than a certain threshold, they can maintain the same thermal history as that of the matrix region owing to the rapid heat conduction effects. Conversely, when these dimensions exceed this threshold, the thermal effects are less pronounced compared to those in the matrix region owing to excessively long heat conduction paths. Similar results are also reported in Ref. 52. Therefore, we believe that our simulation results are reliable, and there is no difference between the thermal histories of the micro-pillars and micro-cantilevers.

Furthermore, we did not observe any distinct traces of σ phase precipitation at the top regions of the micro-pillars (Figs. 4a and 4c), possibly because of the limited resolution of the SEM-based EBSD and EDS methods, which may not be able to discern σ phases with sizes less than 200 nm.

52. M.-J. Kim, S. Yoon, S. Park, H.-J. Jeong, J.-W. Park, K. Kim, J. Jo, T. Heo, S.-T. Hong, S.H. Cho, Y.-K. Kwon, I.-S. Choi, M. Kim, H.N. Han, Elucidating the origin of electroplasticity in metallic materials, *Appl. Mater. Today* **21** (2020) 100874. <https://doi.org/10.1016/j.apmt.2020.100874>.

5. How can the statistical results of Ni, Mn, and Cu content in austenite in Figs. 4-5 be displayed in one graph? Is it a simple sum of the mass percentages of three elements? Their diffusion coefficients should differ.

Response: Thank you for your valuable comments. The microstructures of ferrite-austenite steels are determined by chromium-based (Cr, Mo, Si, Nb, and Ti) and nickel-based (Ni, Mn, Cu, C, and N) alloying elements. The chromium-based alloying elements stabilize the ferrite phase, while the nickel-based ones stabilize the austenite phase. For

the duplex stainless-steel material used in this study, the distribution and diffusion of the nickel-based alloying elements (Ni, Mn, and Cu) affect the austenite distribution and transformation. Hence, in Figs. 4 and 5, we present the simple sum of the contents of Ni, Mn, and Cu, which may better illustrate the influence of the austenite-stabilizing element diffusion on the phase transformation. However, their respective diffusion coefficients were calculated based on the variations in their individual elemental contents (shown in Supplementary Table T1).

6. Is electronic wind the only athermal factor that suppresses precipitated phases? The discussion in this section is too simplistic. There are many athermal effects of current, and it should not be simply assumed that the effect of electron wind inhibits the precipitation of precipitates.

Response: Thank you for your insights and suggestions. Recent studies suggest that the athermal effect mechanism is complex and may involve the EWF, phonon wind force, thermal compressive stress, etc. However, these studies are primarily focused on the effects of electric current on dislocations, i.e., the electroplastic phenomena, in materials. In contrast, in this study, we utilized a fully annealed duplex stainless steel material with no or negligible amount of dislocations. However, this material exhibits compositional differences with Cr concentrated in the α phase and Ni, Mn, and Cu enriched in the γ phase. Hence, the microstructure changes caused by the ECT are more likely related to EWF-induced elemental diffusion (i.e., electromigration) rather than electroplasticity. Thus, the mechanisms of the thermal and athermal effects observed in this study are solely related to Joule heating and the EWF, respectively. Therefore, the final analysis and discussion are solely focused on the EWF.

Additionally, in accordance with your comment, we have provided a more in-depth discussion on how the EWF inhibits the precipitation of the σ phase. The following changes have been made to the revised manuscript:

Page 4, lines 88–95:

Furthermore, a completely annealed duplex stainless-steel (DSS) material was utilized in this study. The microstructure of the DSS contained approximately 50% α -ferrite and 50% γ -austenite phases with compositional differences in phases that Cr concentrated in the α phase and Ni, Mn, and Cu enriched in the γ phase. Moreover, this DSS contained zero or negligible dislocations. Therefore, the selected material simplifies the athermal effect on the EWF (electromigration-related) without showing effects due to dislocations. Accordingly, the analyzed mechanisms of thermal and athermal effects can be attributed to Joule heating and the EWF, respectively.

Pages 18–19, lines 416–446:

Despite having the same thermal history as that of the micro-cantilever, the matrix region does not show σ or other secondary phases (Fig. 5f), indicating that the EWF interrupted the nucleation of the σ phase. Recent studies indicate that the ECT has the

opposite effect on secondary phases or particles. First, the ECT can dissolve secondary phases or particles, and this phenomenon has been applied in electrically assisted solution heat treatments^{63,64}. The ECT can also accelerate the diffusion of alloying elements, thus promoting the nucleation and growth of precipitates, which have now been studied for electrically assisted aging of precipitation-hardened materials^{65–67}. Thus, the ECT demonstrates entirely different effects on secondary phases or particles, and the corresponding key factors depend on the material and applied current conditions. Studies on precipitation show that the promotion of nucleation and growth of the secondary phases often occurs under alternating current treatments with low current densities (less than 20 A/mm²)^{65–67}, which activate the diffusion and movement of alloying elements. These features promote elemental aggregation of secondary phases or particles and finally cause the nucleation and growth of second phases. However, when the current density exceeds a certain threshold, the precipitation of the secondary phases or particles is inhibited⁶⁸, and these phases or particles even dissolve^{63,64} possibly owing to the counteracting effects of the alternating current-induced reciprocating disturbance of the elements and elemental aggregation caused by the thermal equilibrium. Moreover, when the alternating current is switched to a direct current, the secondary phases or particles dissolve, rather than precipitating^{69–72} because the concentration gradient of the elements between the secondary phase or particles and the matrix provides a driving force for diffusion (i.e., electromigration), which disrupts their thermal equilibrium. Additionally, some *In situ* observations show that the secondary phases or particles gradually dissolve and peel off from the side facing the electron wind; this result demonstrates the directional characteristics of the ECT^{70,71}. Therefore, when a direct current with a current density of up to 700 A/mm² is applied, the DSS material used in this study should exhibit secondary-phase dissolution. Unlike the randomness of the atomic diffusion promoted by the Joule heating, the EWF directionally accelerates atomic diffusion, thus inhibiting the aggregation of Cr and precipitation of the σ phases. Consequently, the ECT method is superior to the conventional heat treatment methods for annealing DSS materials, offering an alternative approach for treating DSS materials at 600–1000 °C without precipitating harmful second phases.

7. line 423: previous work is mentioned, but no references are given.

Response: Thank you for your comments. We have added the citation of our previous work in the relevant sentence in the revised manuscript.

Page 23, lines 548–550:

In our previous study, we found a significant enhancement in the mechanical performance and optimization of microstructures after applying 20 pulses of 700 A/mm² for 8 ms on the DSS material used in this study¹⁸.

18. Gu, S. *et al.* Realizing strength–ductility synergy in a lean duplex stainless steel through enhanced TRIP effect via pulsed electric current treatment. *Mater. Sci. Eng. A* **883**, 145534 (2023).

[10.1016/j.msea.2023.145534](https://doi.org/10.1016/j.msea.2023.145534).

8. In the discussion, there is a lack of detailed statements on the mechanism of the influence of athermal and thermal effects on precipitation and phase transition.

Response: Thank you for your comments and suggestions. In the revised manuscript, we have added two subsections in the Discussion part to separately discuss the phase transformation and precipitation caused by the thermal and athermal effects as follows:

Pages 16–18, lines 366–399 (phase transformation):

The metal solid-state phase transformations observed in this study are primarily attributed to thermodynamics due to the significant heat changes induced during the ECT (Fig. 2i and Fig. 3i). Additionally, because of the DSS material used in this study (with differences in chemical composition between the α -ferrite and γ -austenite phases), the solid-state phase transformation is likely correlated with the diffusion of the elements between the phases¹³. The statistical analysis of the EBSD results obtained from the top surfaces of 10 micro-pillars subjected to the Joule heating indicates that the austenite phase content increases by an average of 2.9%, as shown in Fig. 4e. Simultaneously, migration and diffusion of the elements occur within the micromachined structures, as shown in Figs. 4f–g and 5h–i. Therefore, the 2.9% increase in the austenite phase is likely related to thermodynamics and thermomigration of the elements.

Furthermore, we conducted several rapid annealing experiments, in which the samples were immersed in a pre-heated furnace (640 °C) for 1, 2, and 5 min and were then subjected to air cooling. The temperature–time curves of these rapidly annealed samples were estimated using a one-dimensional transient heat conduction model⁵⁶. The results showed that after approximately 1 min of soaking, the sample temperature increased to 635 °C (close to the furnace temperature of 640 °C), as shown in Supplementary Fig. A7. XRD analyses revealed that after 1, 2, and 5 min of rapid annealing, the austenite phase content in the microstructure increased (from 49.5%) to 51.0%, 52.0%, and 52.1%, respectively, with a variation of 1.5%, 2.5%, and 2.6% (Supplementary Fig. A7), respectively. When the annealing time exceeded 2 min, no further increase in the austenite phase content was detected, indicating that it reached a thermal equilibrium state at a temperature of 640 °C. Thus, the pillar top also showed a 2.9% increase in austenite phase content; this result possibly indicated that it reached a similar thermal equilibrium state after 20 times of ECT.

In contrast, in the matrix region (subjected to both Joule heating and the EWF), the austenite phase content increased by approximately 7.6% (based on extensive EBSD and XRD analyses), which was significantly higher than that observed at the top region of the micro-pillars (2.9%; solely affected by the Joule heating). This result indicates that the EWF plays a dominant role in the phase transformation of DSS materials because of the accelerated migration of elements between the phases, especially that of the austenite-stabilizing elements, due to the combined effects of Joule heating and the EWF (Figs. 4f–g and 5h–i), thereby triggering the austenite transformation. Additionally, the EWF may reduce the activation energy and energy barrier of the solid-state phase transformation^{15,57}, resulting in a significant increase in the austenite phase content.

Pages 18–19, lines 402–446 (precipitation):

The DSS annealing within the temperature range 600 °C to 1000 °C often results in the precipitation of detrimental second phases, such as σ phase, χ phase, nitride, and carbide^{58–61}. As observed in this study, the micro-cantilever subjected only to **Joule heating** (reaching a maximum temperature of approximately 640 °C) exhibited significant precipitation of σ phase (Figs. 5e and 5g). The **Joule heating-induced** precipitation of the σ phase (including nucleation and precipitation), which typically contains high levels of Cr and Mo, is attributed to the thermodynamics and elemental diffusion⁶¹, which is random and non-directional. In this study, the precipitation of the σ phase is primarily associated with Cr. **Additionally, previous studies show that a non-isothermal cyclic heat treatment (maximum temperature reaches 600 °C) can accelerate the precipitation of σ phase in DSS materials⁶². In our study, the micromachined DSS structures were also subjected to a similar non-isothermal heat treatment at a maximum temperature of approximately 640 °C for 20 times with an interval of 1 min; these non-isothermal heat-treated micromachined structures showed the precipitation of a large amount of σ phase.**

Despite having the same thermal history as that of the micro-cantilever, the matrix region does not show σ or other secondary phases (Fig. 5f), indicating that the **EFW interrupted the nucleation of the σ phase. Recent studies indicate that the ECT has the opposite effect on secondary phases or particles. First, the ECT can dissolve secondary phases or particles, and this phenomenon has been applied in electrically assisted solution heat treatments^{63,64}. The ECT can also accelerate the diffusion of alloying elements, thus promoting the nucleation and growth of precipitates, which have now been studied for electrically assisted aging of precipitation-hardened materials^{65–67}. Thus, the ECT demonstrates entirely different effects on secondary phases or particles, and the corresponding key factors depend on the material and applied current conditions. Studies on precipitation show that the promotion of nucleation and growth of the secondary phases often occurs under alternating current treatments with low current densities (less than 20 A/mm²)^{65–67}, which activate the diffusion and movement of alloying elements. These features promote elemental aggregation of secondary phases or particles and finally cause the nucleation and growth of second phases. However, when the current density exceeds a certain threshold, the precipitation of the secondary phases or particles is inhibited⁶⁸, and these phases or particles even dissolve^{63,64} possibly owing to the counteracting effects of the alternating current-induced reciprocating disturbance of the elements and elemental aggregation caused by the thermal equilibrium. Moreover, when the alternating current is switched to a direct current, the secondary phases or particles dissolve, rather than precipitating^{69–72} because the concentration gradient of the elements between the secondary phase or particles and the matrix provides a driving force for diffusion (i.e., electromigration), which disrupts their thermal equilibrium. Additionally, some *In situ* observations show that the secondary phases or particles gradually dissolve and peel off from the side facing the electron wind; this result demonstrates the directional characteristics of the ECT^{70,71}. Therefore, when a direct current with a current density of up to 700 A/mm² is applied, the DSS material used in this study should exhibit**

secondary-phase dissolution. Unlike the randomness of the atomic diffusion promoted by the **Joule heating**, the EWF **directionally accelerates** atomic diffusion, **thus inhibiting** the aggregation of Cr and precipitation of the σ phases. Consequently, the ECT method **is superior to the** conventional heat treatment methods **for annealing DSS materials**, offering an alternative **approach** for treating DSS materials at 600–1000 °C without precipitating harmful second phases.

Reviewer #3:

The topic of the paper, utilizing micromachined structures as a way of disentangling the thermal and athermal effects of electric current treatments, is pertinent to the purpose and scope of the journal. The authors' use of pre-micromachined structures is very unique and the method is explained well and backed up by finite element models and experimentation. In terms of the method, this is a well thought out and written paper. The authors also put thought into choosing duplex stainless steel for their validation material so as to look at possible microstructural changes, elemental diffusion, and phase transformations. However, the conclusions that this paper draws on the athermal effect, while possible, are not backed up by statistically significant data. The following are my comments and revision recommendations that would need to be addressed prior to publication:

1. As a methods paper with a set of validation experiments, this is a very nice paper. It is recommended that the authors make clear in the results and discussion sections of the duplex stainless steel experiments that they are potential or exemplary results that could be obtained using this method. The quantitative conclusions that are drawn (length of coherent phase boundaries, diffusion of Cr and Ni, etc.) are based on a single sample and cannot be considered statistically significant.

Response: Thank you for your comments and suggestions. As indicated by you and other reviewers, the experimental results provided in this paper are insufficient for a quantitative analysis of the thermal and athermal effects, as well as for validating the method proposed. Therefore, we have added more experimental results (additional data of nine micro-pillars), especially EBSD results, to enhance the reliability of the experimental results. Additionally, we have provided detailed and insightful discussions and clarifications related to the associated principles in the revised manuscript. For specific experimental results, please refer to Figs. 4 and 5, as well as the relevant sections in the main text and Supplementary Information.

Pages 10–11, lines 213–248:

Fig. 4 displays the *in situ* EBSD and EDS results obtained before and after the pulsed ECT on the pillar top (thermal effect) and matrix (thermal and athermal effects). Figs. 4a and 4b present the band contrast (BC) maps with the γ phase (in pink) at the pillar top and matrix before and after the pulsed ECT. In addition, we fabricated additional nine micro-pillars, which were subsequently characterized using EBSD and EDS techniques. The matrix region was subjected to extensive EBSD and X-ray diffraction (XRD) analyses, as shown in Supplementary Fig. A2–A5. The results indicated that the phase transformation between α and γ phases occurred at the pillar top and matrix. The statistical results of 10 micro-pillars showed that the γ phase content in the pillar top increased by an average of 2.9%, whereas it increased by 7.6% (EBSD: 9.7%, XRD: 5.6%) in the matrix region, as shown in Fig. 4e (detailed in Supplementary Figs. A3–A4). Therefore, the 2.9% phase transformation at the pillar top was attributed to the thermal effect, whereas the phase transformation, which was different from that in the matrix region (4.7%), was ascribed to the athermal effect. This result indicates that in the phase transformation, the contribution of the athermal effect surpasses that of the thermal

effect.

The phase transformation conforms to **body- and face-centered cubic** orientation relationships and **increases** the lengths of the coherent phase boundaries (CPBs, blue lines in Figs. 4a and 4b; Supplementary Fig. A6). **Similar to the phase transformation results, the changes in the CPBs were also more pronounced in the matrix region than at the pillar top: the length of CPBs per unit area in the matrix region increased from 0.10 to 0.17 $\mu\text{m}/\mu\text{m}^2$ and that at the pillar top increased from 0.12 to 0.15 $\mu\text{m}/\mu\text{m}^2$ (Supplementary Fig. A3).**

The distributions of Cr and Ni before and after the pulsed ECT are presented in Figs. 4c and 4d, **and the distributions of Mn and Cu are shown in Supplementary Fig. A5.** In the initial microstructure before the ECT (A and B regions), Cr was enriched in the α phase, while Ni, **Mn, and Cu were** enriched in the γ phase. However, **these alloying elements exhibited a mixing phenomenon after the ECT.** Cr diffused from the α phase to the γ phase, and conversely, Ni, **Mn, and Cu** diffused from the γ phase to the α phase. Figs. 4f and 4g present the changes in the content of Cr within the α phase and **those of Ni, Mn, and Cu within the γ phase; these values were obtained from 40 sampling points before and after the ECT (Supplementary Fig. A5).** Because the matrix regions show the same thermal history as that of the micromachined structures but experience an additional athermal effect, changes in the elemental content in the micromachined regions may be attributed to the thermal effect, while differences of the elemental content between micromachined structures and matrix should be ascribed to the athermal effect. Thus, these results reveal that the thermal and athermal effects decrease the Cr content (α phase) by 0.19% and 0.06%, respectively, as well as decrease the Ni, Mn, and Cu contents (γ phase) by 0.21% and 0.14%, respectively.

Figs. 4–5, and Figs. A2–A7 in Supplementary Information:

Please refer to the revised manuscript.

2. There appear to be error bars on some of the datapoints in Figures 4e,f,g, and 5h,i but it is not clear how those were obtained. Additionally, the overlap in the error bars, for example in Figure 4g and 5h, show the elemental content % decrease attributed to the thermal and thermal + athermal, respectively, are within the original error of the starting value. While these trends may be correct, additional samples or larger area scans would need to be performed to provide statistical significance and reliability to the conclusions that are drawn.

Response: Thank you for your comments and suggestions. In the revised manuscript, we have added the data of additional nine pillars and have thus analyzed the results obtained from a total of 10 micropillars to improve the reliability of experimental results. For improving the statistics in Figs. 4f–g and Figs. 5h–i, we have increased the sampling number from 3–6 to 40 and 20, respectively. For the corresponding modifications, please refer to our response to the previous comment 1.

3. In lines 261, the authors state they observe the emergence of numerous hemline-like structures. Could the authors elaborate on what they mean by “hemline-like structures”? From the figure, it appears the arrows are pointing to grain precipitates.

Response: Thank you for your comments. The hemline-like structure is a ring- or crescent-shaped structure that appears around the γ phase after the ECT. It is related to the diffusion of the austenite-stabilizing elements, which triggers the precipitation of small α -grains, as shown in the schematic below.

Schematic of the hemline-like structure formation.

The corresponding modifications made in the revised manuscript are indicated below:

Page 14, lines 294–300:

Additionally, we observed the emergence of numerous hemline-like (ring- or crescent-shaped) structures around the γ phase after the ECT in the matrix region, which was subjected to both thermal and athermal effects (B') (as indicated by the yellow arrows in Fig. 6a). These hemline-like structures were small α -grains precipitated ahead of the γ phase. However, these special structures were not observed in the micromachined structures, which were subjected only to the thermal effect, and TEM observations also confirmed this phenomenon (Fig. 6b).

4. In lines 46-47, the authors state that the athermal effect primarily involves electron wind force. There is much in the literature that refutes an electron wind force and while research shows that an athermal effect is occurring, the cause of the athermal effect is still much debated and could likely be caused by different mechanisms depending on the material system of interest. It is recommended the authors make clear that a leading hypothesis is an electron wind effect unless the authors cite specific research for duplex stainless steel.

Response: Thank you for your insightful and valuable comments. Indeed, many recent studies suggest that the athermal effect originates from complex mechanisms, which still remain a subject of debate. These studies were primarily focused on the effects of electric current on dislocations, i.e., the electroplastic phenomena in materials. In contrast, in our study, we utilized a fully annealed duplex stainless steel material with no or very few dislocations. However, this material exhibits compositional differences with Cr concentrated in the α phase and Ni, Mn, and Cu enriched in the γ phase. Hence, the microstructure changes caused by the ECT are more likely related to EWF-induced

elemental diffusion (i.e., electromigration) rather than electroplasticity. Thus, the mechanisms of the thermal and athermal effects observed in this study are solely related to Joule heating and the EWF, respectively. The following changes have been made in the revised manuscript:

Page 2, lines 45–56:

The thermal effect primarily involves Joule heating, whereas the athermal effect follows a complex mechanism. Theoretical and experimental evidences pin the electron wind force (EWF) as the origin of the athermal effect in electromigration (atomic diffusion)^{29–31}. However, the principles of electroplasticity (dislocation motion behavior) have not yet been clarified. Initially, Troitskii et al. attributed the motion of electro-induced dislocations to the EWF^{3,4}. Some recent studies have also shown that the EWF plays a decisive role in dislocation motion and reconfiguration^{32,33}. However, the quantified value of the EWF is insufficient to drive dislocation movement, and other causes, such as phonon wind force and thermal gradient or compressive stress, have been proposed^{26,27}. Recent studies suggest that because different materials exhibit different electroplastic phenomena, the mechanisms of the athermal effect underlying this phenomenon might not have a unified explanation^{26–28}.

Page 4, lines 88–95:

Furthermore, a completely annealed duplex stainless-steel (DSS) material was utilized in this study. The microstructure of the DSS contained approximately 50% α -ferrite and 50% γ -austenite phases with compositional differences in phases that Cr concentrated in the α phase and Ni, Mn, and Cu enriched in the γ phase. Moreover, this DSS contained zero or negligible dislocations. Therefore, the selected material simplifies the athermal effect on the EWF (electromigration-related) without showing effects due to dislocations. Accordingly, the analyzed mechanisms of thermal and athermal effects can be attributed to Joule heating and the EWF, respectively.

5. In the analysis of the thermal effect of the matrix versus the cantilever, the authors state in lines 184-185 that they experience the same thermal effect. This would be true for bulk Joule heating but have the authors considered possible localized joule heating at grain boundaries/dislocations that may occur with electric field versus without and how that may affect the temperature?

Response: Thank you for your comments. Recent studies indicate a disagreement related to the issue of localized heating at dislocations or grain boundaries. Some studies suggest that localized heating does not occur at grain boundaries owing to the rapid heat conduction of the material⁵². In contrast, others argue that localized thermal zones exist around dislocations and promote the electroplasticity effect in materials^{R5}. However, these studies are primarily based on simulation analyses and lack experimental evidences.

We believe that the micro defects (dislocations or grain boundaries) do not undergo

obvious local heating. Our rationale is similar to that of the aforementioned studies (Ref. [52]), which report that rapid heat conduction dissipates local heating rapidly, resulting in a uniform thermal distribution. In the present study, we also leveraged this characteristic (rapid heat conduction) to maintain the micromachined structures with the same thermal history as that of the matrix. However, this structure had a size limitation (the TCP length of the micromachined structures needed to be smaller than 50 μm in this study); otherwise, significant thermal gradient or localized heating would have occurred (the temperature of the micromachined structures would be significantly lower than that of the matrix). However, the scale of the grain boundaries and dislocations was significantly smaller than this limit (50 μm). Therefore, we believe that there is no obvious local heating around the grain boundaries or dislocations in the material subjected to the current conditions used in this study.

Pages 14–15, lines 317–321:

In addition, we believe that no obvious local heating occurs at the grain boundaries⁵² to facilitate the diffusion of elements owing to the rapid heat conduction during the ECT. Accordingly, the element diffusion between α and γ phases is quantitatively investigated based on the uniform spatial heat distribution in the following discussion.

52. M.-J. Kim, S. Yoon, S. Park, H.-J. Jeong, J.-W. Park, K. Kim, J. Jo, T. Heo, S.-T. Hong, S.H. Cho, Y.-K. Kwon, I.-S. Choi, M. Kim, H.N. Han, Elucidating the origin of electroplasticity in metallic materials, *Appl. Mater. Today* **21** (2020) 100874. <https://doi.org/10.1016/j.apmt.2020.100874>.

R5. Z. Xu, X. Li, R. Zhang, J. Ma, D. Qiu, L. Peng, The effect of electric current on dislocation activity in pure aluminum: A 3D discrete dislocation dynamics study, *Int. J. Plasticity* **171** (2023) 103826. <https://doi.org/10.1016/j.ijplas.2023.103826>.

6. In terms of adding statistical significance to the EBSD results for the pillars, the authors may want to consider taking a much larger scan area prior to performing the FIB of the pillar. This would 1) provide analysis of many more grains and 2) show that the FIB work itself doesn't alter the microstructure. Following that, if changes are only seen in the micropillar and nowhere else, the results would be more significant without the burden of many more samples being tested.

Response: We appreciate your kind advice and suggestions. Following the comments of the reviewers, we conducted additional experiments (including increasing the number of micropillars, scanning larger areas of the EBSD images, and conducting XRD experiments) to provide credible experimental data for revealing the thermal and athermal effects. Moreover, these distinct experimental results and conclusions also support the feasibility and effectiveness of the method proposed in this study (the pre-micromachined structure utilization method). Therefore, in the revised manuscript, we have added details of these additional experiments as well as in-depth discussions on our experimental results. For detailed modifications, please refer to the revised manuscript's Figs. 4 and 5, as well as the discussion section.

Figs. 4–5, and Figs. A2–A7 in Supplementary Information:
Please refer to the revised manuscript.

REVIEWERS' COMMENTS

Reviewer #1 (Remarks to the Author):

After the revision, the manuscript appears to have achieved a higher level of completeness. However, it should be noted that in both the introduction and the main body, the concept of atomic diffusion enhancement due to electric current has been equated with electron wind force (EWF), which is not entirely accurate. Strictly speaking, electron wind force represents one hypothesis explaining atomic diffusion enhancement by electric current, and there exists a diversity of opinions among researchers regarding this matter. The primary objective of this study was to observe athermal and thermal effects at the microscale using micropillars. It did not entail a physical or theoretical approach to the atomic diffusion enhancement in electroplasticity. Therefore, to enhance clarity and avoid confusion, it is recommended to revise the sections where atomic diffusion enhancement is equated with electron wind force (EWF)

Reviewer #2 (Remarks to the Author):

In general, the authors have satisfactorily responded to the reviewer's comments. However, some grammatical errors still remain in the paper, so the authors should have this paper proofread by a native English speaker.

Reviewer #3 (Remarks to the Author):

The authors have addressed all previous reservations and it is now my recommendation that this manuscript be published. This reviewer thanks the authors for their responses and additional information that provides clarity to the manuscript.

Responses to Reviewers

Title: Micromachined Structures Decoupling Joule Heating and Electron Wind Force

Dear Reviewers,

We are grateful to the reviewers for their valuable suggests and insightful comments on our manuscript again. Our point-by-point responses to the comments of the reviewers have been appended below, and the changes made in the revised manuscript have been highlighted in **red color** for ease of identification.

Reviewer #1:

After the revision, the manuscript appears to have achieved a higher level of completeness. However, it should be noted that in both the introduction and the main body, the concept of atomic diffusion enhancement due to electric current has been equated with electron wind force (EWF), which is not entirely accurate. Strictly speaking, electron wind force represents one hypothesis explaining atomic diffusion enhancement by electric current, and there exists a diversity of opinions among researchers regarding this matter. The primary objective of this study was to observe athermal and thermal effects at the microscale using micropillars. It did not entail a physical or theoretical approach to the atomic diffusion enhancement in electroplasticity. Therefore, to enhance clarity and avoid confusion, it is recommended to revise the sections where atomic diffusion enhancement is equated with electron wind force (EWF).

Response: Thank you for your comments and suggestions. We share your concerns and have added several clarifications throughout the manuscript, indicating that the electron wind force (EWF) is currently considered the primary driver of element diffusion, supported by numerous theoretical and experimental studies. However, other athermal effects may also contribute to element diffusion. Due to the lack of research evidence and theoretical basis, this study will not discuss the impact of athermal effects other than the EWF on element diffusion. Our revisions are as follows.

Page 4, lines 93–96:

Therefore, the selected material simplifies the athermal effect on the EWF (**element diffusion-related**), and **whether the athermal effect other than the EWF influences element diffusion and phase transitions will not be discussed in this study due to the lack of experimental and theoretical evidence.**

Page 15, lines 334–337:

Additionally, other athermal effects apart from the EWF may also contribute to the diffusion of elements. However, due to the lack of experimental and theoretical evidence, these effects will not be discussed in this section.

Page 18, lines 404–405:

Due to the lack of evidence, whether the athermal effect other than the EWF affects the α - γ phase transition will not be discussed in this study.

Reviewer #2:

In general, the authors have satisfactorily responded to the reviewer's comments. However, some grammatical errors still remain in the paper, so the authors should have this paper proofread by a native English speaker.

Response: Thank you very much for your comments. We have utilized a professional English editing service to proofread our manuscript. All grammatical revisions have been highlighted in red in the revised manuscript. The authors would like to once again thank you for your highly valuable comments and suggestions this time and last time.

Reviewer #3:

The authors have addressed all previous reservations and it is now my recommendation that this manuscript be published. This reviewer thanks the authors for their responses and additional information that provides clarity to the manuscript.

Response: The authors appreciate your previous constructive feedback and suggestions, which were crucial in improving the quality of our manuscript. We are grateful for your contribution.